# Fundamental Convergence Analysis of Sharpness-Aware Minimization

**Pham Duy Khanh**
Ho Chi Minh City University of Education
khanhpd@hcmue.edu.vn

**Hoang-Chau Luong**
VNU-HCM University of Science
lhchau20@apcs.fitus.edu.vn

**Boris S. Mordukhovich**
Wayne State University
boris@math.wayne.edu

**Dat Ba Tran**
Wayne State University
tranbadat@wayne.edu

## Abstract

The paper investigates the fundamental convergence properties of Sharpness-Aware Minimization (SAM), a recently proposed gradient-based optimization method [Foret et al., 2021] that significantly improves the generalization of deep neural networks. The convergence properties including the stationarity of accumulation points, the convergence of the sequence of gradients to the origin, the sequence of function values to the optimal value, and the sequence of iterates to the optimal solution are established for the method. The universality of the provided convergence analysis based on inexact gradient descent frameworks [Khanh et al., 2023b] allows its extensions to efficient normalized versions of SAM such as F-SAM [Li et al., 2024], VaSSO [Li and Giannakis, 2023], RSAM [Liu et al., 2022], and to the unnormalized versions of SAM such as USAM [Andriushchenko and Flammarion, 2022]. Numerical experiments are conducted on classification tasks using deep learning models to confirm the practical aspects of our analysis.

## 1   Introduction

This paper concentrates on optimization methods for solving the standard optimization problem

$$\text{minimize} \quad f(x) \text{ subject to } x \in \mathbb{R}^n, \tag{1}$$

where $f : \mathbb{R}^n \to \mathbb{R}$ is a continuously differentiable ($\mathcal{C}^1$-smooth) function. We study the convergence behavior of the gradient-based optimization algorithm *Sharpness-Aware Minimization* [Foret et al., 2021] together with its efficient practical variants [Liu et al., 2022, Li and Giannakis, 2023, Andriushchenko and Flammarion, 2022]. Given an initial point $x^1 \in \mathbb{R}^n$, the original iterative procedure of SAM is designed as follows

$$x^{k+1} = x^k - t\nabla f\left(x^k + \rho\frac{\nabla f(x^k)}{\|\nabla f(x^k)\|}\right) \tag{2}$$

for all $k \in \mathbb{N}$, where $t > 0$ and $\rho > 0$ are respectively the *stepsize* (in other words, the learning rate) and *perturbation radius*. The main motivation for the construction algorithm is that by making the backward step $x^k + \rho\frac{\nabla f(x^k)}{\|\nabla f(x^k)\|}$, it avoids minimizers with large sharpness, which is usually poor for the generalization of deep neural networks as shown in Keskar et al. [2017].

### 1.1   Lack of convergence properties for SAM due to constant stepsize

The consistently high efficiency of SAM has driven a recent surge of interest in the analysis of the method. The convergence analysis of SAM is now a primary focus on its theoretical understanding

38th Conference on Neural Information Processing Systems (NeurIPS 2024).

with several works being developed recently (e.g., Ahn et al. [2024], Andriushchenko and Flammarion [2022], Dai et al. [2023], Si and Yun [2023]). However, none of the aforementioned studies have addressed the fundamental convergence properties of SAM, which are outlined below where the stationary accumulation point in (2) means that every accumulation point $\bar{x}$ of the iterative sequence $\{x^k\}$ satisfies the condition $\nabla f(\bar{x}) = 0$.

| | |
|---|---|
| (1) | $\liminf\limits_{k\to\infty} \left\|\nabla f(x^k)\right\| = 0$ |
| (2) | stationary accumulation point |
| (3) | $\lim\limits_{k\to\infty} \left\|\nabla f(x^k)\right\| = 0$ |
| (4) | $\lim\limits_{k\to\infty} f(x^k) = f(\bar{x})$ with $\nabla f(\bar{x}) = 0$ |
| (5) | $\{x^k\}$ converges to some $\bar{x}$ with $\nabla f(\bar{x}) = 0$ |

Table 1: Fundamental convergence properties of smooth optimization methods

The relationship between the properties above is summarized as follows:

$$(1) \overset{\{\|x^k\|\}\not\to\infty}{\Longleftarrow} (2) \Longleftarrow (3) \Longleftarrow (5) \Longrightarrow (4).$$

The aforementioned convergence properties are standard and are analyzed by various smooth optimization methods including gradient descent-type methods, Newton-type methods, and their accelerated versions together with nonsmooth optimization methods under the usage of subgradients. The readers are referred to Bertsekas [2016], Nesterov [2018], Polyak [1987] and the references therein for those results. The following recent publications have considered various types of convergence rates for the sequences generated by SAM as outlined below:

(i) Dai et al. [2023, Theorem 1]

$$f(x^k) - f^* \le (1 - t\mu(2 - Lt))^k (f(x^0) - f^*) + \frac{tL^2\rho^2}{2\mu(2 - Lt)}$$

where $L$ is the Lipschitz constant of $\nabla f$, and where $\mu$ is the constant of strong convexity constant for $f$.

(ii) Si and Yun [2023, Theorems 3.3, 3.4]

$$\frac{1}{k}\sum_{i=1}^{k}\left\|\nabla f(x^i)\right\|^2 = \mathcal{O}\left(\frac{1}{k} + \frac{1}{\sqrt{k}}\right) \quad \text{and} \quad \frac{1}{k}\sum_{i=1}^{k}\left\|\nabla f(x^i)\right\|^2 = \mathcal{O}\left(\frac{1}{k}\right) + L^2\rho^2,$$

where $L$ is the Lipschitz constant of $\nabla f$. We emphasize that none of the results mentioned above achieve any fundamental convergence properties listed in Table 1. The estimation in (i) only gives us the convergence of the function value sequence to a value close to the optimal one, not the convergence to exactly the optimal value. Additionally, it is evident that the results in (ii) do not imply the convergence of $\{\nabla f(x^k)\}$ to 0. To the best of our knowledge, the only work concerning the fundamental convergence listed in Table 1 is Andriushchenko and Flammarion [2022]. However, the method analyzed in that paper is unnormalized SAM (USAM), a variant of SAM with the norm being removed in the iterative procedure (2c). Recently, Dai et al. [2023] suggested that USAM has different effects in comparison with SAM in both practical and theoretical situations, and thus, they should be addressed separately. This observation once again highlights the necessity for a fundamental convergence analysis of SAM and its normalized variants.

Note that, using exactly the iterative procedure (2), SAM does not achieve the convergence for either $\{x^k\}$, or $\{f(x^k)\}$, or $\{\nabla f(x^k)\}$ to the optimal solution, the optimal value, and the origin, respectively. It is illustrated by Example 3.1 below dealing with quadratic functions. This calls for the necessity of employing an alternative stepsize rule for SAM. Scrutinizing the numerical experiments conducted for SAM and its variants (e.g., Foret et al. [2021, Subsection C1], Li and Giannakis [2023, Subsection 4.1]), we can observe that in fact the constant stepsize rule is not a preferred strategy. Instead, the cosine stepsize scheduler from Loshchilov and Hutter [2016], designed to decay to zero and then restart after each fixed cycle, emerges as a more favorable approach. This observation

motivates us to analyze the method under diminishing stepsize, which is standard and employed in many optimization methods including the classical gradient descent methods together with its incremental and stochastic counterparts [Bertsekas and Tsitsiklis, 2000]. Diminishing step sizes also converge to zero as the number of iterations increases, a condition satisfied by the practical cosine step size scheduler in each cycle.

## 1.2 Our Contributions

**Convergence of SAM and normalized variants**

We establish fundamental convergence properties of SAM in various settings. In the convex case, we consider the perturbation radii to be variable and bounded. This analysis encompasses the practical implementation of SAM with a constant perturbation radius. The results in this case are summarized in Table 2.

| Classes of function | Results |
|---|---|
| General setting | $\liminf \nabla f(x^k) = 0$ |
| Bounded minimizer set | stationary accumulation point |
| Unique minimizer | $\{x^k\}$ is convergent |

Table 2: Convergence properties of SAM for convex functions in Theorem 3.2

In the nonconvex case, we present a unified convergence analysis framework that can be applied to most variants of SAM, particularly recent efficient developments such as VaSSO [Li and Giannakis, 2023], F-SAM [Li et al., 2024], and RSAM [Liu et al., 2022]. We observe that all these methods can be viewed as inexact gradient descent (IGD) methods with absolute error. This version of IGD has not been previously considered, and its convergence analysis is significantly more complex than the one in Khanh et al. [2023b, 2024a, 2023a, 2024b], as the function value sequence generated by the new algorithm may not be decreasing. This disrupts the convergence framework for monotone function value sequences used in the aforementioned works. To address this challenge, we adapt the analysis for algorithms with nonmonotone function value sequences from Li et al. [2023], which was originally developed for random reshuffling algorithms, a context entirely different from ours.

We establish the convergence of IGD with absolute error when the perturbation radii decrease at an *arbitrarily slow rate*. Although the analysis of this general framework does not theoretically cover the case of a constant perturbation radius, it poses no issues for the practical implementation of these methods, as discussed in Remark 3.6. A summary of our results in the nonconvex case is provided in the first part of Table 3.

**Convergence of USAM and unnormalized variants**

Our last theoretical contribution in this paper involves a refined convergence analysis of USAM in Andriushchenko and Flammarion [2022]. In the general setting, we address functions satisfying the $L$-descent condition (4), which is even weaker than the Lipschitz continuity of $\nabla f$ as considered in Andriushchenko and Flammarion [2022]. The summary of the convergence analysis for USAM is given in the second part of Table 3.

As will be discussed in Remark G.4, our convergence properties for USAM use weaker assumptions and cover a broader range of applications in comparison with those analyzed in [Andriushchenko and Flammarion, 2022]. Furthermore, the universality of the conducted analysis allows us to verify all the convergence properties for the extragradient method [Korpelevich, 1976] that has been recently applied in Lin et al. [2020] to large-batch training in deep learning.

| SAM and normalized variants | | USAM and unnormalized variants | |
|---|---|---|---|
| **Classes of functions** | **Results** | **Classes of functions** | **Results** |
| General setting | $\lim \nabla f(x^k) = 0$ | General setting | stationary accumulation point |
| General setting | $\lim f(x^k) = f^*$ | General setting | $\lim f(x^k) = f^*$ |
| KL property | $\{x^k\}$ is convergent | $\nabla f$ is Lipschitz | $\lim \nabla f(x^k) = 0$ |
| | | KL property | $\{x^k\}$ is convergent |

Table 3: Convergence properties of SAM together with normalized variants (Corollary 3.5, Appendix D), and USAM together with unnormalized variants (Theorem 4.2)

## 1.3 Importance of Our Work

Our work develops, for the first time in the literature, a fairly comprehensive analysis of the fundamental convergence properties of SAM and its variants. The developed approach addresses general frameworks while being based on the analysis of the newly proposed inexact gradient descent methods. Such an approach can be applied in various other circumstances and provides useful insights into the convergence understanding of not only SAM and related methods but also many other numerical methods in smooth, nonsmooth, and derivative-free optimization.

## 1.4 Related Works

**Variants of SAM**. There have been several publications considering some variants to improve the performance of SAM. Namely, Kwon et al. [2021] developed the Adaptive Sharpness-Aware Minimization (ASAM) method by employing the concept of normalization operator. Du et al. [2022] proposed the Efficient Sharpness-Aware Minimization (ESAM) algorithm by combining stochastic weight perturbation and sharpness-sensitive data selection techniques. Liu et al. [2022] proposed a novel Random Smoothing-Based SAM method called RSAM that improves the approximation quality in the backward step. Quite recently, Li and Giannakis [2023] proposed another approach called Variance Suppressed Sharpness-aware Optimization (VaSSO), which perturbed the backward step by incorporating information from the previous iterations. As Li et al. [2024] identified noise in stochastic gradient as a crucial factor in enhancing SAM's performance, they proposed Friendly Sharpness-Aware Minimization (F-SAM) which perturbed the backward step by extracting noise from the difference between the stochastic gradient and the expected gradient at the current step. Two efficient algorithms, AE-SAM and AE-LookSAM, are also proposed in Jiang et al. [2023], by employing adaptive policy based on the loss landscape geometry.

**Theoretical Understanding of SAM**. Despite the success of SAM in practice, a theoretical understanding of SAM was absent until several recent works. Barlett et al. [2023] analyzed the convergence behavior of SAM for convex quadratic objectives, showing that for most random initialization, it converges to a cycle that oscillates between either side of the minimum in the direction with the largest curvature. Ahn et al. [2024] introduces the notion of $\varepsilon$-approximate flat minima and investigates the iteration complexity of optimization methods to find such approximate flat minima. As discussed in Subsection 1.1, Dai et al. [2023] considers the convergence of SAM with constant stepsize and constant perturbation radius for convex and strongly convex functions, showing that the sequence of iterates stays in a neighborhood of the global minimizer while Si and Yun [2023] considered the properties of the gradient sequence generated by SAM in different settings.

**Theoretical Understanding of USAM**. This method was first proposed by Andriushchenko and Flammarion [2022] with fundamental convergence properties being analyzed under different settings of convex and nonconvex and optimization. Analysis of USAM was further conducted in Behdin and Mazumder [2023] for a linear regression model, and in Agarwala and Dauphin [2023] for a quadratic regression model. Detailed comparison between SAM and USAM, which indicates that they exhibit different behaviors, was presented in the two recent studies by Compagnoni et al. [2023] and Dai et al. [2023]. During the final preparation of the paper, we observed that the convergence of USAM can also be deduced from Mangasarian and Solodov [1994], though under some additional assumptions, including the boundedness of the gradient sequence.

## 2 Preliminaries

First we recall some notions and notations frequently used in the paper. All our considerations are given in the space $\mathbb{R}^n$ with the Euclidean norm $\| \cdot \|$. As always, $\mathbb{N} := \{1, 2, \ldots\}$ signifies the collections of natural numbers. The symbol $x^k \xrightarrow{J} \bar{x}$ means that $x^k \to \bar{x}$ as $k \to \infty$ with $k \in J \subset \mathbb{N}$. Recall that $\bar{x}$ is a *stationary point* of a $\mathcal{C}^1$-smooth function $f : \mathbb{R}^n \to \mathbb{R}$ if $\nabla f(\bar{x}) = 0$. A function $f : \mathbb{R}^n \to \mathbb{R}$ is said to posses a Lipschitz continuous gradient with the uniform constant $L > 0$, or equivalently it belongs to the class $\mathcal{C}^{1,L}$, if we have the estimate

$$\|\nabla f(x) - \nabla f(y)\| \le L \|x - y\| \quad \text{for all} \ \ x, y \in \mathbb{R}^n. \tag{3}$$

This class of function enjoys the following property called the *L-descent condition* (see, e.g., Izmailov and Solodov [2014, Lemma A.11] and Bertsekas [2016, Lemma A.24]):

$$f(y) \leq f(x) + \langle \nabla f(x), y - x \rangle + \frac{L}{2} \|y - x\|^2 \tag{4}$$

for all $x, y \in \mathbb{R}^n$. Conditions (3) and (4) are equivalent to each other when $f$ is convex, while the equivalence fails otherwise. A major class of functions satisfying the $L$-descent condition but not having the Lipschitz continuous gradient is given by Khanh et al. [2023b, Section 2] as $f(x) := \frac{1}{2} \langle Ax, x \rangle + \langle b, x \rangle + c - h(x)$, where $A$ is an $n \times n$ matrix, $b \in \mathbb{R}^n$, $c \in \mathbb{R}$ and $h : \mathbb{R}^n \to \mathbb{R}$ is a smooth convex function whose gradient is not Lipschitz continuous. There are also circumstances where a function has a Lipschitz continuous gradient and satisfies the descent condition at the same time, but the Lipschitz constant is larger than the one in the descent condition.

Our convergence analysis of the methods presented in the subsequent sections benefits from the *Kurdyka-Łojasiewicz* $(KL)$ *property* taken from Attouch et al. [2010].

**Definition 2.1** (Kurdyka-Łojasiewicz property). We say that a smooth function $f : \mathbb{R}^n \to \mathbb{R}$ enjoys the *KL property* at $\bar{x} \in \text{dom} \, \partial f$ if there exist $\eta \in (0, \infty]$, a neighborhood $U$ of $\bar{x}$, and a desingularizing concave continuous function $\varphi : [0, \eta) \to [0, \infty)$ such that $\varphi(0) = 0$, $\varphi$ is $\mathcal{C}^1$-smooth on $(0, \eta)$, $\varphi' > 0$ on $(0, \eta)$, and for all $x \in U$ with $0 < f(x) - f(\bar{x}) < \eta$, we have

$$\varphi'(f(x) - f(\bar{x})) \|\nabla f(x)\| \geq 1. \tag{5}$$

*Remark* 2.2. It has been realized that the KL property is satisfied in broad settings. In particular, it holds at every *nonstationary point* of $f$; see Attouch et al. [2010, Lemma 2.1 and Remark 3.2(b)]. Furthermore, it is proved at the seminal paper [Łojasiewicz, 1965] that any analytic function $f : \mathbb{R}^n \to \mathbb{R}$ satisfies the KL property on $\mathbb{R}^n$ with $\varphi(t) = M t^{1-q}$ for some $q \in [0, 1)$. Typical functions that satisfy the KL property are *semi-algebraic* functions and in general, functions *definable in o-minimal structures*; see Attouch et al. [2010, 2013], Kurdyka [1998].

We utilize the following assumption on the desingularizing function in Definition 2.1, which is employed in Li et al. [2023]. The satisfaction of this assumption for a general class of desingularizing functions is discussed in Remark G.1.

**Assumption 2.3.** There is some $C > 0$ such that whenever $x, y \in (0, \eta)$ with $x + y < \eta$, it holds that

$$C[\varphi'(x + y)]^{-1} \leq (\varphi'(x))^{-1} + (\varphi'(y))^{-1}.$$

## 3 SAM and normalized variants

### 3.1 Convex case

We begin this subsection with an example that illustrates SAM's inability to achieve the convergence of the sequence of iterates to an optimal solution of strongly convex quadratic functions by using a constant stepsize. This emphasizes the necessity of avoiding this type of stepsize in our subsequent analysis.

*Example* 3.1 (SAM with constant stepsize and constant perturbation radius does not converge). Let the sequence $\{x^k\}$ be generated by SAM in (2) applied to the strongly convex quadratic function $f(x) = \frac{1}{2} \langle Ax, x \rangle - \langle b, x \rangle$, where $A$ is an $n \times n$ symmetric, positive-definite matrix and $b \in \mathbb{R}^n$. Then for any fixed small constant perturbation radius and for some small constant stepsize together with an initial point close to the solution, the sequence $\{x^k\}$ generated by this algorithm does not converge to the optimal solution.

The details of the above example are presented in Appendix A.1. Figure 1 gives an empirical illustration for Example 3.1. The graph shows that, while the sequence generated by GD converges to 0, the one generated by SAM gets stuck at a different point.

As the constant stepsize does not guarantee the convergence of SAM, we consider another well-known stepsize called diminishing (see (7)). The following result provides the convergence properties of SAM in the convex case for that type of stepsize.

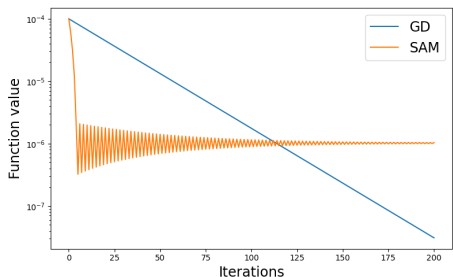

Figure 1: SAM with constant stepsize does not converge to optimal solution

**Theorem 3.2.** *Let $f : \mathbb{R}^n \to \mathbb{R}$ be a smooth convex function whose gradient is Lipschitz continuous with constant $L > 0$. Given any initial point $x^1 \in \mathbb{R}^n$, let $\{x^k\}$ be generated by the SAM method with the iterative procedure*

$$x^{k+1} = x^k - t_k \nabla f \left( x^k + \rho_k \frac{\nabla f(x^k)}{\|\nabla f(x^k)\|} \right) \tag{6}$$

*for all $k \in \mathbb{N}$ with nonnegative stepsizes and perturbation radii satisfying the conditions*

$$\sum_{k=1}^{\infty} t_k^2 < \infty, \ \sum_{k=1}^{\infty} t_k = \infty, \ \sup_{k \in \mathbb{N}} \rho_k < \infty. \tag{7}$$

*Assume that $\nabla f(x^k) \neq 0$ for all $k \in \mathbb{N}$ and that $\inf_{k \in \mathbb{N}} f(x^k) > -\infty$. Then the following assertions hold:*

**(i)** $\liminf_{k \to \infty} \nabla f(x^k) = 0.$

**(ii)** *If $f$ has a nonempty bounded level set, then $\{x^k\}$ is bounded, every accumulation point of $\{x^k\}$ is a global minimizer of $f$, and $\{f(x^k)\}$ converges to the optimal value of $f$. If in addition $f$ has a unique minimizer, then the sequence $\{x^k\}$ converges to that minimizer.*

Due to the space limit, the proof of the theorem is presented in Appendix C.1.

### 3.2 Nonconvex case

In this subsection, we study the convergence of several versions of SAM from the perspective of the inexact gradient descent methods.

---

**Algorithm 1** Inexact Gradient Descent (IGD) Methods

---

*Step 0.* Choose some initial point $x^0 \in \mathbb{R}^n$, sequence of errors $\{\varepsilon_k\} \subset (0, \infty)$, and sequence of stepsizes $\{t_k\} \subset (0, \infty)$. For $k = 1, 2, \ldots$, do the following

*Step 1.* Set $x^{k+1} = x^k - t_k g^k$ with $\|g^k - \nabla f(x^k)\| \leq \varepsilon_k$.

---

This algorithm is motivated by while being different from the Inexact Gradient Descent methods proposed in [Khanh et al., 2023a,b, 2024b,a]. The latter constructions consider relative errors in gradient calculation, while Algorithm 1 uses the absolute ones. This approach is particularly suitable for the constructions of SAM and its normalized variants. The convergence properties of Algorithm 1 are presented in the next theorem.

**Theorem 3.3.** *Let $f : \mathbb{R}^n \to \mathbb{R}$ be a smooth function whose gradient is Lipschitz continuous with some constant $L > 0$, and let $\{x^k\}$ be generated by the IGD method in Algorithm 1 with stepsizes and errors satisfying the conditions*

$$\sum_{k=1}^{\infty} t_k = \infty, \ t_k \downarrow 0, \sum_{k=1}^{\infty} t_k \varepsilon_k < \infty, \ \limsup \varepsilon_k < 2. \tag{8}$$

*Assume that* $\inf_{k \in \mathbb{N}} f(x^k) > -\infty$. *Then the following convergence properties hold:*

**(i)** $\nabla f(x^k) \to 0$, *and thus every accumulation point of* $\{x^k\}$ *is stationary for* $f$.

**(ii)** *If* $\bar{x}$ *is an accumulation point of the sequence* $\{x^k\}$, *then* $f(x^k) \to f(\bar{x})$.

**(iii)** *Suppose that* $f$ *satisfies the KL property at some accumulation point* $\bar{x}$ *of* $\{x^k\}$ *with the desingularizing function* $\varphi$ *satisfying Assumption 2.3. Assume in addition that*

$$\sum_{k=1}^{\infty} t_k \left( \varphi' \left( \sum_{i=k}^{\infty} t_k \varepsilon_k \right) \right)^{-1} < \infty, \tag{9}$$

*and that* $f(x^k) > f(\bar{x})$ *for sufficiently large* $k \in \mathbb{N}$. *Then* $x^k \to \bar{x}$ *as* $k \to \infty$. *In particular, if* $\bar{x}$ *is a global minimizer of* $f$, *then either* $f(x^k) = f(\bar{x})$ *for some* $k \in \mathbb{N}$, *or* $x^k \to \bar{x}$.

The proof of the theorem is presented in Appendix C.2. The demonstration that condition (9) is satisfied when $\varphi(t) = Mt^{1-q}$ with some $M > 0$ and $q \in (0, 1)$, and when $t_k = \frac{1}{k}$ and $\varepsilon_k = \frac{1}{k^p}$ with $p \geq 0$ for all $k \in \mathbb{N}$, is provided in Remark G.3.

The next example discusses the necessity of the last two conditions in (8) in the convergence analysis of IGD while demonstrating that employing a constant error leads to the convergence to a nonstationary point of the method.

*Example* 3.4 (IGD with constant error converges to a nonstationary point). Let $f : \mathbb{R} \to \mathbb{R}$ be defined by $f(x) = x^2$ for $x \in \mathbb{R}$. Given a perturbation radius $\rho > 0$ and an initial point $x^1 > \rho$, consider the iterative sequence

$$x^{k+1} = x^k - 2t_k \left( x^k - \rho \frac{x^k}{|x^k|} \right) \quad \text{for } k \in \mathbb{N}, \tag{10}$$

where $\{t_k\} \subset [0, 1/2], t_k \downarrow 0$, and $\sum_{k=1}^{\infty} t_k = \infty$. This algorithm is in fact the IGD applied to $f$ with $g^k = \nabla f \left( x^k - \rho \frac{f'(x^k)}{|f'(x^k)|} \right)$. Then $\{x^k\}$ converges to $\rho$, which is not a stationary point of $f$.

The details of the example are presented in Appendix A.2. We now propose a general framework that encompasses SAM and all of its normalized variants including RSAM [Liu et al., 2022], VaSSO [Li and Giannakis, 2023] and F-SAM [Li et al., 2024]. Due to the page limit, we refer readers to Appendix D for the detailed constructions of those methods. Remark D.1 in Appendix D also shows that all of these methods are special cases of Algorithm 1a, and thus all the convergence properties presented in Theorem 3.3 follow.

---

**Algorithm 1a** General framework for normalized variants of SAM

---

*Step* 0. Choose $x^1 \in \mathbb{R}^n$, $\{\rho_k\}, \{t_k\} \subset (0, \infty)$, and $\{d^k\} \subset \mathbb{R}^n \setminus \{0\}$. For $k \geq 1$ do the following:

*Step* 1. Set $x^{k+1} = x^k - t_k \nabla f \left( x^k + \rho_k \frac{d^k}{\|d^k\|} \right)$.

---

**Corollary 3.5.** *Let* $f : \mathbb{R}^n \to \mathbb{R}$ *be a* $\mathcal{C}^{1,L}$ *function, and let* $\{x^k\}$ *be generated by Algorithm 1a with the parameters*

$$\sum_{k=1}^{\infty} t_k = \infty, \ t_k \downarrow 0, \sum_{k=1}^{\infty} t_k \rho_k < \infty, \ \limsup \rho_k < \frac{2}{L}. \tag{11}$$

*Assume that* $\inf_{k \in \mathbb{N}} f(x^k) > -\infty$. *Then all convergence properties presented in Theorem 3.3 hold.*

The proof of this result is presented in Appendix C.5.

*Remark* 3.6. Note that the conditions in (11) do not pose any obstacles to the implementation of a constant perturbation radius for SAM in practical circumstances. This is due to the fact that a possible selection of $t_k$ and $\rho_k$ satisfying (11) is $t_k = \frac{1}{k}$ and $\rho_k = \frac{C}{k^{0.001}}$ for all $k \in \mathbb{N}$ (almost constant), where $C > 0$. Then the initial perturbation radius is $C$, while after $C$ million iterations, it remains greater than $0.99C$. This phenomenon is also confirmed by numerical experiments in Appendix E on nonconvex functions. The numerical results show that SAM with almost constant radii $\rho_k = \frac{C}{k^p}$ has a similar convergence behavior to SAM with a constant radius $\rho = C$. As SAM with a constant perturbation radius has sufficient empirical evidence for its efficiency in Foret et al. [2021], this also supports the practicality of our almost constant perturbation radii.

# 4 USAM and unnormalized variants

In this section, we study the convergence of various versions of USAM from the perspective of the following Inexact Gradient Descent method with relative errors.

---

**Algorithm 2** IGDr

---

*Step* 0. Choose some $x^0 \in \mathbb{R}^n, \nu \geq 0$, and $\{t_k\} \subset [0, \infty)$. For $k = 1, 2, \ldots$, do the following:

*Step* 1. Set $x^{k+1} = x^k - t_k g^k$, where $\left\| g^k - \nabla f(x^k) \right\| \leq \nu \left\| \nabla f(x^k) \right\|$.

---

This algorithm was initially introduced in Khanh et al. [2023b] in a different form, considering a different selection of error. The form of IGDr closest to Algorithm 2 was established in Khanh et al. [2024a] and then further studied in Khanh et al. [2024a, 2023a, 2024b]. In this paper, we extend the analysis of the method to a general stepsize rule covering both constant and diminishing cases, which was not considered in Khanh et al. [2024a].

**Theorem 4.1.** *Let $f : \mathbb{R}^n \to \mathbb{R}$ be a smooth function satisfying the descent condition for some constant $L > 0$, and let $\{x^k\}$ be the sequence generated by Algorithm 2 with the relative error $\nu \in [0, 1)$, and the stepsizes satisfying*

$$\sum_{k=1}^{\infty} t_k = \infty \ \ and \ \ t_k \in \left[0, \frac{2 - 2\nu - \delta}{L(1 + \nu)^2}\right] \tag{12}$$

*for sufficiently large $k \in \mathbb{N}$ and for some $\delta > 0$. Then either $f(x^k) \to -\infty$, or we have the assertions:*

**(i)** *Every accumulation point of $\{x^k\}$ is a stationary point of the cost function $f$.*

**(ii)** *If the sequence $\{x^k\}$ has any accumulation point $\bar{x}$, then $f(x^k) \downarrow f(\bar{x})$.*

**(iii)** *If $f \in \mathcal{C}^{1,L}$, then $\nabla f(x^k) \to 0$.*

**(iv)** *If $f$ satisfies the KL property at some accumulation point $\bar{x}$ of $f$, then $\{x^k\} \to \bar{x}$.*

**(v)** *Assume in addition to (iv) that the stepsizes are bounded away from 0, and the KL property in (iv) holds with the desingularizing function $\varphi(t) = Mt^{1-q}$ with $M > 0$ and $q \in (0, 1)$. Then either $\{x^k\}$ stops finitely at a stationary point, or the following convergence rates are achieved:*

- *If $q = 1/2$, then $\{x^k\}$, $\{\nabla f(x^k)\}$, $\{f(x^k)\}$ converge linearly as $k \to \infty$ to $\bar{x}$, $0$, and $f(\bar{x})$.*

- *If $q \in (1/2, 1)$, then*

$$\left\| x^k - \bar{x} \right\| = \mathcal{O}\left(k^{-\frac{1-q}{2q-1}}\right), \ \left\| \nabla f(x^k) \right\| = \mathcal{O}\left(k^{-\frac{1-q}{2q-1}}\right), \ f(x^k) - f(\bar{x}) = \mathcal{O}\left(k^{-\frac{2-2q}{2q-1}}\right).$$

Although the ideas for proving this result is similar to the one given in Khanh et al. [2024a], we do provide the full proof in the Appendix C.3 for the convenience of the readers. We now show that using this approach, we derive more complete convergence results for USAM in Andriushchenko and Flammarion [2022] and also the extragradient method by Korpelevich [1976], Lin et al. [2020].

---

**Algorithm 2a** [Andriushchenko and Flammarion, 2022] Unnormalized Sharpness-Aware Minimization (USAM)

---

*Step* 0. Choose $x^0 \in \mathbb{R}^n$, $\{\rho_k\} \subset [0, \infty)$, and $\{t_k\} \subset [0, \infty)$. For $k = 1, 2, \ldots$, do the following:

*Step* 1. Set $x^{k+1} = x^k - t_k \nabla f(x^k + \rho_k \nabla f(x^k))$.

---

**Algorithm 2b** [Korpelevich, 1976] Extragradient Method

---

*Step* 0. Choose $x^0 \in \mathbb{R}^n$, $\{\rho_k\} \subset [0, \infty)$, and $\{t_k\} \subset [0, \infty)$. For $k = 1, 2, \ldots$, do the following:

*Step* 1. Set $x^{k+1} = x^k - t_k \nabla f(x^k - \rho_k \nabla f(x^k))$.

---

We are ready now to derive convergence of the two algorithms above. The proof of the theorem is given in Appendix C.4

**Theorem 4.2.** *Let* $f : \mathbb{R}^n \to \mathbb{R}$ *be a* $\mathcal{C}^1$-*smooth function satisfying the descent condition with some constant* $L > 0$. *Let* $\{x^k\}$ *be the sequence generated by either Algorithm 2a, or by Algorithm 2b with* $\rho_k \leq \frac{\nu}{L}$ *for some* $\nu \in [0, 1)$ *and with the stepsize satisfying* (12). *Then all the convergence properties in Theorem 4.1 hold.*

## 5  Numerical Experiments

To validate the practical aspect of our theory, this section compares the performance of SAM employing constant and diminishing stepsizes in image classification tasks. All the experiments are conducted on a computer with NVIDIA RTX 3090 GPU. The three types of diminishing stepsizes considered in the numerical experiments are $\eta_1/n$ (Diminish 1), $\eta_1/n^{0.5001}$ (Diminish 2), and $\eta_1/m \log m$ (Diminish 3), where $\eta_1$ is the initial stepsize, $n$ represents the number of epochs performed, and $m = \lfloor n/5 \rfloor + 2$. The constant stepsize in SAM is selected through a grid search over $\{0.1, 0.01, 0.001\}$ to ensure a fair comparison with the diminishing ones. The algorithms are tested on two widely used image datasets: CIFAR-10 [Krizhevsky et al., 2009] and CIFAR-100 [Krizhevsky et al., 2009].

**CIFAR-10**. We train well-known deep neural networks including ResNet18 [He et al., 2016], ResNet34 [He et al., 2016], and WideResNet28-10 [Zagoruyko and Komodakis, 2016] on this dataset by using 10% of the training set as a validation set. Basic transformations, including random crop, random horizontal flip, normalization, and cutout [DeVries and Taylor, 2017], are employed for data augmentation. All the models are trained by using SAM with SGD Momentum as the base optimizer for 200 epochs and a batch size of 128. This base optimizer is also used in the original paper [Foret et al., 2021] and in the recent works on SAM [Ahn et al., 2024, Li and Giannakis, 2023]. Following the approach by Foret et al. [2021], we set the initial stepsize to 0.1, momentum to 0.9, the $\ell_2$-regularization parameter to 0.001, and the perturbation radius $\rho$ to 0.05. Setting the perturbation radius to be a constant here does not go against our theory, since by Remark 3.6, SAM with a constant radius and our almost constant radius have the same numerical behavior. We also conduct the numerical experiment with an almost constant radius and got the same results. Therefore, for simplicity of presentation, a constant perturbation radius is chosen. The algorithm with the highest accuracy, corresponding to the best performance, is highlighted in bold. The results in Table 5 report the mean and 95% confidence interval across the three independent runs. The training loss in several tests is presented in Figure 2.

**CIFAR-100**. The training configurations for this dataset are similar to CIFAR10. The accuracy results are presented in Table 5, while the training loss results are illustrated in Figure 2.

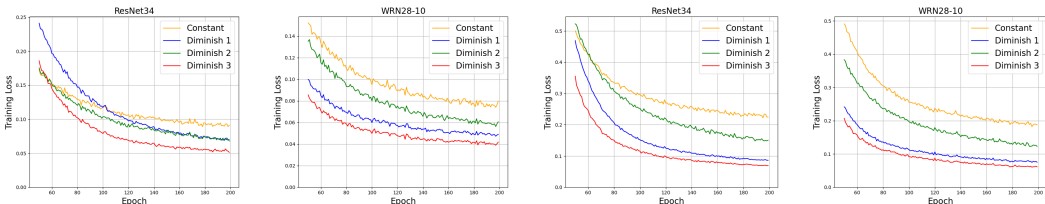

Figure 2: Training loss on CIFAR-10 (first two graphs) and CIFAR-100 (last two graphs)

| | CIFAR-10 | | | CIFAR-100 | | |
|---|---|---|---|---|---|---|
| Model | ResNet18 | ResNet34 | WRN28-10 | ResNet18 | ResNet34 | WRN28-10 |
| Constant | $94.10 \pm 0.27$ | $94.38 \pm 0.47$ | $95.33 \pm 0.23$ | $71.77 \pm 0.26$ | $72.49 \pm 0.23$ | $74.63 \pm 0.84$ |
| Diminish 1 | $93.95 \pm 0.34$ | $93.94 \pm 0.40$ | $95.18 \pm 0.03$ | $74.43 \pm 0.12$ | $73.99 \pm 0.70$ | $78.59 \pm 0.03$ |
| Diminish 2 | $94.60 \pm 0.09$ | $\mathbf{95.09 \pm 0.16}$ | $95.75 \pm 0.23$ | $73.40 \pm 0.24$ | $74.44 \pm 0.89$ | $77.04 \pm 0.23$ |
| Diminish 3 | $\mathbf{94.75 \pm 0.20}$ | $94.47 \pm 0.08$ | $\mathbf{95.88 \pm 0.10}$ | $\mathbf{75.65 \pm 0.44}$ | $\mathbf{74.92 \pm 0.76}$ | $\mathbf{79.70 \pm 0.12}$ |

Table 4: Test accuracy on CIFAR-10 and CIFAR-100

The results on CIFAR-10 and CIFAR-100 indicate that SAM with **Diminish 3** stepsize usually achieves the best performance in both accuracy and training loss among all tested stepsizes. In all the architectures used in the experiment, the results consistently show that diminishing stepsizes outperform constant stepsizes in terms of both accuracy and training loss measures. Additional numerical results on a larger data set and without momentum can be found in Appendix F.

## 6 Discusison

### 6.1 Conclusion

In this paper, we provide a fundamental convergence analysis of SAM and its normalized variants together with a refined convergence analysis of USAM and its unnormalized variants. Our analysis is conducted in deterministic settings under standard assumptions that cover a broad range of applications of the methods in both convex and nonconvex optimization. The conducted analysis is universal and thus can be applied in different contexts other than SAM and its variants. The performed numerical experiments show that our analysis matches the efficient implementations of SAM and its variants that are used in practice.

### 6.2 Limitations

Our analysis is only conducted in deterministic settings, which leaves the stochastic and random reshuffling developments to our future research. The analysis of SAM coupling with momentum methods is also not considered in this paper. Another limitation pertains to numerical experiments, where only SAM was tested on three different architectures of deep learning.

## Acknowledgment

Pham Duy Khanh, Research of this author is funded by the Ministry of Education and Training Research Funding under the project B2024-SPS-07. Boris S. Mordukhovich, Research of this author was partly supported by the US National Science Foundation under grants DMS-1808978 and DMS-2204519, by the Australian Research Council under grant DP-190100555, and by Project 111 of China under grant D21024. Dat Ba Tran, Research of this author was partly supported by the US National Science Foundation under grants DMS-1808978 and DMS-2204519.

The authors would like to thank Professor Mikhail V. Solodov for his fruitful discussions on the convergence of variants of SAM.

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

# Contents

# A Counterexamples illustrating the Insufficiency of Fundamental Convergence Properties

## A.1 Proof of Example 3.1

*Proof.* Since $f(x) = \frac{1}{2}\langle Ax, x\rangle - \langle b, x\rangle$, the gradient of $f$ and the optimal solution are given by

$$\nabla f(x) = Ax - b \quad \text{and} \quad x^* = A^{-1}b.$$

Let $\lambda_{\min}, \lambda_{\max} > 0$ be the minimum, maximum eigenvalues of $A$, respectively and assume that

$$t \in \left(\frac{1}{\lambda_{\min}} - \frac{1}{\lambda_{\max} + \lambda_{\min}}, \frac{1}{\lambda_{\min}}\right), \; \rho > 0, \; \text{and} \; 0 < \|x^1 - x^*\| < \frac{t\rho\lambda_{\min}}{1 - t\lambda_{\min}}. \tag{13}$$

The iterative procedure of (6) can be written as

$$x^{k+1} = x^k - t\nabla f\left(x^k + \rho\frac{\nabla f(x^k)}{\|\nabla f(x^k)\|}\right) = x^k - t\left[A\left(x^k + \rho\frac{Ax^k - b}{\|Ax^k - b\|}\right) - b\right]. \tag{14}$$

Then $\{x^k\}$ satisfies the inequalities

$$0 < \|x^k - A^{-1}b\| < \frac{t\rho\lambda_{\min}}{1 - t\lambda_{\min}} \quad \text{for all} \; k \in \mathbb{N}. \tag{15}$$

It is obvious that (15) holds for $k = 1$. Assuming that this condition holds for any $k \in \mathbb{N}$, let us show that it holds for $k + 1$. We deduce from the iterative update (14) that

$$\begin{aligned}
\|x^{k+1} - A^{-1}b\| &= \left\|x^k - t\left[A\left(x^k + \rho\frac{Ax^k - b}{\|Ax^k - b\|}\right) - b\right] - A^{-1}b\right\| \\
&= \left\|(I - tA)(x^k - A^{-1}b) - t\rho\frac{A(Ax^k - b)}{\|Ax^k - b)\|}\right\| \\
&\geq \left\|t\rho\frac{A(Ax^k - b)}{\|Ax^k - b\|}\right\| - \|(I - tA)(x^k - A^{-1}b)\| \\
&\geq t\rho\lambda_{\min} - (1 - t\lambda_{\min})\|x^k - A^{-1}b\| > 0. \tag{16}
\end{aligned}$$

In addition, we get

$$\begin{aligned}
\|x^{k+1} - A^{-1}b\| &\leq \left\|t\rho\frac{A(Ax^k - b)}{\|Ax^k - b\|}\right\| + \|(I - tA)(x^k - A^{-1}b)\| \\
&\leq t\rho\lambda_{\max} + (1 - t\lambda_{\min})\|x^k - A^{-1}b\| \\
&\leq t\rho\lambda_{\max} + t\rho\lambda_{\min} < t\rho\frac{\lambda_{\min}}{1 - t\lambda_{\min}}
\end{aligned}$$

where the last inequality follows from $t > \frac{1}{\lambda_{\min}} - \frac{1}{\lambda_{\max} + \lambda_{\min}}$ from (13). Thus, (15) is verified. It follows from (16) that $x^k \nrightarrow x^*$. $\qquad\square$

## A.2 Proof of Example 3.4

*Proof.* Observe that $x^k > \rho$ for all $k \in \mathbb{N}$. Indeed, this follows from $x^1 > \rho$, $t_k < 1/2$, and

$$x^{k+1} - \rho = x^k - \rho - 2t_k(x^k - \rho) = (1 - 2t_k)(x^k - \rho).$$

In addition, we readily get

$$0 \leq x^{k+1} - \rho = (1 - 2t_k)(x^k - \rho) = \ldots = (x^1 - \rho)\prod_{i=1}^{k}(1 - 2t_i). \tag{17}$$

Furthermore, deduce from $\sum_{k=1}^{\infty} 2t_k = \infty$ that $\prod_{k=1}^{\infty}(1 - 2t_k) = 0$. Indeed, we have

$$0 \leq \prod_{k=1}^{\infty}(1 - 2t_k) \leq \frac{1}{\prod_{k=1}^{\infty}(1 + 2t_k)} \leq \frac{1}{1 + \sum_{k=1}^{\infty} 2t_k} = 0.$$

This tells us by (17) and the classical squeeze theorem that $x^k \to \rho$ as $k \to \infty$. $\qquad\square$

## B   Auxiliary Results for Convergence Analysis

We first establish the new three sequences lemma, which is crucial in the analysis of both SAM, USAM, and their variants.

**Lemma B.1** (three sequences lemma). Let $\{\alpha_k\}, \{\beta_k\}, \{\gamma_k\}$ be sequences of nonnegative numbers satisfying the conditions

$$\alpha_{k+1} - \alpha_k \leq \beta_k \alpha_k + \gamma_k \ \text{ for sufficient large } \ k \in \mathbb{N}, \tag{a}$$

$$\{\beta_k\} \text{ is bounded}, \sum_{k=1}^{\infty} \beta_k = \infty, \ \sum_{k=1}^{\infty} \gamma_k < \infty, \ \text{ and } \ \sum_{k=1}^{\infty} \beta_k \alpha_k^2 < \infty. \tag{b}$$

Then we have that $\alpha_k \to 0$ as $k \to \infty$.

*Proof.* First we show that $\liminf_{k\to\infty} \alpha_k = 0$. Supposing the contrary gives us some $\delta > 0$ and $N \in \mathbb{N}$ such that $\alpha_k \geq \delta$ for all $k \geq N$. Combining this with the second and the third condition in (b) yields

$$\infty > \sum_{k=N}^{\infty} \beta_k \alpha_k^2 \geq \delta^2 \sum_{k=N}^{\infty} \beta_k = \infty,$$

which is a contradiction verifying the claim. Let us now show that in fact $\lim_{k\to\infty} \alpha_k = 0$. Indeed, by the boundedness of $\{\beta_k\}$ define $\bar{\beta} := \sup_{k\in\mathbb{N}} \beta_k$ and deduce from (a) that there exists $K \in \mathbb{N}$ such that

$$\alpha_{k+1} - \alpha_k \leq \beta_k \alpha_k + \gamma_k \text{ for all } k \geq K. \tag{18}$$

Pick $\varepsilon > 0$ and find by $\liminf_{k\to\infty} \alpha_k = 0$ and the two last conditions in (b) some $K_\varepsilon \in \mathbb{N}$ with $K_\varepsilon \geq K, \alpha_{K_\varepsilon} \leq \varepsilon$,

$$\sum_{k=K_\varepsilon}^{\infty} \gamma_k < \frac{\varepsilon}{3}, \sum_{k=K_\varepsilon}^{\infty} \beta_k \alpha_k^2 < \frac{\varepsilon^2}{3}, \ \text{ and } \ \bar{\beta}\beta_k \alpha_k^2 \leq \frac{\varepsilon^2}{9} \ \text{ as } \ k \geq K_\varepsilon. \tag{19}$$

It suffices to show that $\alpha_k \leq 2\varepsilon$ for all $k \geq K_\varepsilon$. Fix $k \geq K_\varepsilon$ and observe that for $\alpha_k \leq \varepsilon$ the desired inequality is obviously satisfied. If $\alpha_k > \varepsilon$, we use $\alpha_{K_\varepsilon} \leq \varepsilon$ and find some $k' < k$ such that $k' \geq K_\varepsilon$ and

$$\alpha_{k'} \leq \varepsilon \ \text{ and } \ \alpha_i > \varepsilon \ \text{ for } \ i = k, k-1, \ldots, k'+1.$$

Then we deduce from (18) and (19) that

$$\begin{aligned}
\alpha_k - \alpha_{k'} = \sum_{i=k'}^{k-1} (\alpha_{i+1} - \alpha_i) &\leq \sum_{i=k'}^{k-1} (\beta_i \alpha_i + \gamma_i) \\
&= \sum_{i=k'+1}^{k} \beta_i \alpha_i + \sum_{i=k'}^{k-1} \gamma_i + \beta_{k'} \alpha_{k'} \\
&\leq \frac{1}{\varepsilon} \sum_{i=k'+1}^{k} \beta_i \alpha_i^2 + \sum_{i=k'}^{k-1} \gamma_i + \sqrt{\beta_{k'}} \sqrt{\beta_{k'}} \alpha_{k'} \\
&\leq \frac{1}{\varepsilon} \sum_{i=K_\varepsilon}^{\infty} \beta_i \alpha_i^2 + \sum_{i=K_\varepsilon}^{\infty} \gamma_i + \sqrt{\bar{\beta}\beta_{k'}\alpha_{k'}^2} \\
&\leq \frac{1}{\varepsilon} \frac{\varepsilon^2}{3} + \frac{\varepsilon}{3} + \frac{\varepsilon}{3} = \varepsilon.
\end{aligned}$$

As a consequence, we arrive at the estimate

$$\alpha_k = \alpha_{k'} + \alpha_k - \alpha_{k'} \leq \varepsilon + \varepsilon = 2\varepsilon \ \text{ for all } \ k \geq K_\varepsilon,$$

which verifies that $\alpha_k \to 0$ as $k \to \infty$ sand thus completes the proof of the lemma. $\qquad\square$

Next we recall some auxiliary results from Khanh et al. [2023b].

**Lemma B.2.** *Let $\{x^k\}$ and $\{d^k\}$ be sequences in $\mathbb{R}^n$ satisfying the condition*

$$\sum_{k=1}^{\infty} \|x^{k+1} - x^k\| \cdot \|d^k\| < \infty.$$

*If $\bar{x}$ is an accumulation point of the sequence $\{x^k\}$ and $0$ is an accumulation points of the sequence $\{d^k\}$, then there exists an infinite set $J \subset \mathbb{N}$ such that we have*

$$x^k \xrightarrow{J} \bar{x} \quad and \quad d^k \xrightarrow{J} 0. \tag{20}$$

**Proposition B.3.** *Let $f : \mathbb{R}^n \to \mathbb{R}$ be a $\mathcal{C}^1$-smooth function, and let the sequence $\{x^k\} \subset \mathbb{R}^n$ satisfy the conditions:*

**(H1)** *(primary descent condition). There exists $\sigma > 0$ such that for sufficiently large $k \in \mathbb{N}$ we have*

$$f(x^k) - f(x^{k+1}) \geq \sigma \|\nabla f(x^k)\| \cdot \|x^{k+1} - x^k\|.$$

**(H2)** *(complementary descent condition). For sufficiently large $k \in \mathbb{N}$, we have*

$$\left[f(x^{k+1}) = f(x^k)\right] \Longrightarrow [x^{k+1} = x^k].$$

*If $\bar{x}$ is an accumulation point of $\{x^k\}$ and $f$ satisfies the KL property at $\bar{x}$, then $x^k \to \bar{x}$ as $k \to \infty$.*

When the sequence under consideration is generated by a linesearch method and satisfies some conditions stronger than (H1) and (H2) in Proposition B.3, its convergence rates are established in Khanh et al. [2023b, Proposition 2.4] under the KL property with $\psi(t) = Mt^{1-q}$ as given below.

**Proposition B.4.** *Let $f : \mathbb{R}^n \to \mathbb{R}$ be a $\mathcal{C}^1$-smooth function, and let the sequences $\{x^k\} \subset \mathbb{R}^n, \{\tau_k\} \subset [0, \infty), \{d^k\} \subset \mathbb{R}^n$ satisfy the iterative condition $x^{k+1} = x^k + \tau_k d^k$ for all $k \in \mathbb{N}$. Assume that for all sufficiently large $k \in \mathbb{N}$ we have $x^{k+1} \neq x^k$ and the estimates*

$$f(x^k) - f(x^{k+1}) \geq \beta \tau_k \|d^k\|^2 \quad and \quad \|\nabla f(x^k)\| \leq \alpha \|d^k\|, \tag{21}$$

*where $\alpha, \beta > 0$. Suppose in addition that the sequence $\{\tau_k\}$ is bounded away from $0$ (i.e., there is some $\bar{\tau} > 0$ such that $\tau_k \geq \bar{\tau}$ for large $k \in \mathbb{N}$), that $\bar{x}$ is an accumulation point of $\{x^k\}$, and that $f$ satisfies the KL property at $\bar{x}$ with $\psi(t) = Mt^{1-q}$ for some $M > 0$ and $q \in (0, 1)$. Then the following convergence rates are guaranteed:*

**(i)** *If $q \in (0, 1/2]$, then the sequence $\{x^k\}$ converges linearly to $\bar{x}$.*

**(ii)** *If $q \in (1/2, 1)$, then we have the estimate*

$$\|x^k - \bar{x}\| = \mathcal{O}\left(k^{-\frac{1-q}{2q-1}}\right).$$

Yet another auxiliary result needed below is as follows.

**Proposition B.5.** *Let $f : \mathbb{R}^n \to \mathbb{R}$ be a $\mathcal{C}^1$-smooth function satisfying the descent condition (4) with some constant $L > 0$. Let $\{x^k\}$ be a sequence in $\mathbb{R}^n$ that converges to $\bar{x}$, and let $\alpha > 0$ be such that*

$$\alpha \|\nabla f(x^k)\|^2 \leq f(x^k) - f(x^{k+1}) \text{ for sufficiently large } \in \mathbb{N}. \tag{22}$$

*Consider the following convergence rates of $\{x^k\}$ :*

**(i)** $x^k \to \bar{x}$ *linearly.*

**(ii)** $\|x^k - \bar{x}\| = \mathcal{O}(m(k))$, *where $m(k) \downarrow 0$ as $k \to \infty$.*

Then (i) *ensures the linear convergences of $f(x^k)$ to $f(\bar{x})$, and $\nabla f(x^k)$ to $0$, while* (ii) *yields $|f(x^k) - f(\bar{x})| = \mathcal{O}(m^2(k))$ and $\|\nabla f(x^k)\| = \mathcal{O}(m(k))$ as $k \to \infty$.*

*Proof.* Condition (22) tells us that there exists some $N \in \mathbb{N}$ such that $f(x^{k+1}) \le f(x^k)$ for all $k \ge \mathbb{N}$. As $x^k \to \bar{x}$, we deduce that $f(x^k) \to f(\bar{x})$ with $f(x^k) \ge f(\bar{x})$ for $k \ge N$. Letting $k \to \infty$ in (22) and using the squeeze theorem together with the convergence of $\{x^k\}$ to $\bar{x}$ and the continuity of $\nabla f$ lead us to $\nabla f(\bar{x}) = 0$. It follows from the descent condition of $f$ with constant $L > 0$ and from (4) that

$$0 \le f(x^k) - f(\bar{x}) \le \langle \nabla f(\bar{x}), x^k - \bar{x} \rangle + \frac{L}{2} \left\| x^k - \bar{x} \right\|^2 = \frac{L}{2} \left\| x^k - \bar{x} \right\|^2.$$

This verifies the desired convergence rates of $\{f(x^k)\}$. Employing finally (22) and $f(x^{k+1}) \ge f(\bar{x})$, we also get that

$$\alpha \left\| \nabla f(x^k) \right\|^2 \le f(x^k) - f(\bar{x}) \text{ for all } k \ge N.$$

This immediately gives us the desired convergence rates for $\{\nabla f(x^k)\}$ and completes the proof. $\square$

## C  Proof of Convergence Results

### C.1  Proof of Theorem 3.2

*Proof.* To verify (i) first, for any $k \in \mathbb{N}$ define $g^k := \nabla f \left( x^k + \rho_k \frac{\nabla f(x^k)}{\|\nabla f(x^k)\|} \right)$ and get

$$\left\| g^k - \nabla f(x^k) \right\| = \left\| \nabla f \left( x^k + \rho_k \frac{\nabla f(x^k)}{\|\nabla f(x^k)\|} \right) - \nabla f(x^k) \right\|$$

$$\le L \left\| x^k + \rho_k \frac{\nabla f(x^k)}{\|\nabla f(x^k)\|} - x^k \right\| = L\rho_k \le L\rho, \tag{23}$$

where $\rho := \sup_{k \in \mathbb{N}} \rho_k$. Using the monotonicity of $\nabla f$ due to the convexity of $f$ ensures that

$$\langle g^k, \nabla f(x^k) \rangle = \left\langle \nabla f \left( x^k + \rho_k \frac{\nabla f(x^k)}{\|\nabla f(x^k)\|} \right) - \nabla f(x^k), \nabla f(x^k) \right\rangle + \left\| \nabla f(x^k) \right\|^2$$

$$= \frac{\|\nabla f(x^k)\|}{\rho_k} \left\langle \nabla f \left( x^k + \rho_k \frac{\nabla f(x^k)}{\|\nabla f(x^k)\|} \right) - \nabla f(x^k), \rho_k \frac{\nabla f(x^k)}{\|\nabla f(x^k)\|} \right\rangle + \left\| \nabla f(x^k) \right\|^2 \ge \left\| \nabla f(x^k) \right\|^2.$$
$$\tag{24}$$

With the definition of $g^k$, the iterative procedure (6) can also be rewritten as $x^{k+1} = x^k - t_k g^k$ for all $k \in \mathbb{N}$. The first condition in (7) yields $t_k \downarrow 0$, which gives us some $K \in \mathbb{N}$ such that $Lt_k < 1$ for all $k \ge K$. Take some such $k$. Since $\nabla f$ is Lipschitz continuous with constant $L > 0$, it follows from the descent condition in (4) and the estimates in (23), (24) that

$$f(x^{k+1}) \le f(x^k) + \langle \nabla f(x^k), x^{k+1} - x^k \rangle + \frac{L}{2} \left\| x^{k+1} - x^k \right\|^2$$

$$= f(x^k) - t_k \langle \nabla f(x^k), g^k \rangle + \frac{Lt_k^2}{2} \left\| g^k \right\|^2$$

$$= f(x^k) - t_k(1 - Lt_k) \langle \nabla f(x^k), g^k \rangle + \frac{Lt_k^2}{2} \left( \left\| g^k - \nabla f(x^k) \right\|^2 - \left\| \nabla f(x^k) \right\|^2 \right)$$

$$\le f(x^k) - t_k(1 - Lt_k) \left\| \nabla f(x^k) \right\|^2 - \frac{Lt_k^2}{2} \left\| \nabla f(x^k) \right\|^2 + \frac{L^3 t_k^2 \rho^2}{2}$$

$$= f(x^k) - \frac{t_k}{2} (2 - Lt_k) \left\| \nabla f(x^k) \right\|^2 + \frac{L^3 t_k^2 \rho^2}{2}$$

$$\le f(x^k) - \frac{t_k}{2} \left\| \nabla f(x^k) \right\|^2 + \frac{L^3 t_k^2 \rho^2}{2}. \tag{25}$$

Rearranging the terms above gives us the estimate

$$\frac{t_k}{2} \left\| \nabla f(x^k) \right\|^2 \le f(x^k) - f(x^{k+1}) + \frac{L^3 t_k^2 \rho^2}{2}. \tag{26}$$

Select any $M > K$, define $S := \frac{L^3 \rho^2}{2} \sum_{k=1}^{\infty} t_k^2 < \infty$, and get by taking into account $\inf_{k \in \mathbb{N}} f(x^k) > -\infty$ that

$$\frac{1}{2} \sum_{k=K}^{M} t_k \left\| \nabla f(x^k) \right\|^2 \leq \sum_{k=K}^{M} \left( f(x^k) - f(x^{k+1}) \right) + \sum_{k=K}^{M} \frac{L^3 t_k^2 \rho^2}{2}$$
$$\leq f(x^K) - f(x^{M+1}) + S$$
$$\leq f(x^K) - \inf_{k \in \mathbb{N}} f(x^k) + S.$$

Letting $M \to \infty$ yields $\sum_{k=K}^{\infty} t_k \left\| \nabla f(x^k) \right\|^2 < \infty$. Let us now show that $\liminf \left\| \nabla f(x^k) \right\| = 0$. Supposing the contrary gives us $\varepsilon > 0$ and $N \geq K$ such that $\left\| \nabla f(x^k) \right\| \geq \varepsilon$ for all $k \geq N$, which tells us that

$$\infty > \sum_{k=N}^{\infty} t_k \left\| \nabla f(x^k) \right\|^2 \geq \varepsilon^2 \sum_{k=N}^{\infty} t_k = \infty.$$

This clearly contradicts the second condition in (7) and this justifies (i).

To verify (ii), define $u_k := \frac{L^3 \rho^2}{2} \sum_{i=k}^{\infty} t_i^2$ for all $k \in \mathbb{N}$ and deduce from the first condition in (7) that $u_k \downarrow 0$ as $k \to \infty$. With the usage of $\{u_k\}$, estimate (26) is written as

$$f(x^{k+1}) + u_{k+1} \leq f(x^k) + u_k - \frac{t_k}{2} \left\| \nabla f(x^k) \right\|^2 \quad \text{for all } k \geq K,$$

which means that $\left\{ f(x^k) + u_k \right\}_{k \geq K}$ is nonincreasing. It follows from $\inf_{k \in \mathbb{N}} f(x^k) > -\infty$ and $u_k \downarrow 0$ that $\left\{ f(x^k) + u_k \right\}$ is convergent, which means that the sequence $\left\{ f(x^k) \right\}$ is convergent as well. Assume now $f$ has some nonempty and bounded level set. Then every level set of $f$ is bounded by Ruszczynski [2006, Exercise 2.12]. By (26), we get that

$$f(x^{k+1}) \leq f(x^k) + \frac{L^3 \rho^2}{2} t_k^2 - \frac{t_k}{2} \left\| \nabla f(x^k) \right\|^2 \leq f(x^k) + \frac{L^3 \rho^2}{2} t_k^2 \quad \text{for all } k \geq K.$$

Proceeding by induction leads us to

$$f(x^{k+1}) \leq f(x^1) + \frac{L^3 \rho^2}{2} \sum_{i=1}^{k} t_i^2 \leq f(x^1) + S \quad \text{for all } k \geq K,$$

which means that $x^{k+1} \in \left\{ x \in \mathbb{R}^n \mid f(x) \leq f(x^1) + S \right\}$ for all $k \geq K$ and thus justifies the boundedness of $\left\{ x^k \right\}$.

Taking $\liminf \left\| \nabla f(x^k) \right\| = 0$ into account gives us an infinite set $J \subset \mathbb{N}$ such that $\left\| \nabla f(x^k) \right\| \xrightarrow{J} 0$. As $\left\{ x^k \right\}$ is bounded, the sequence $\left\{ x^k \right\}_{k \in J}$ is also bounded, which gives us another infinite set $I \subset J$ and $\bar{x} \in \mathbb{R}^n$ such that $x^k \xrightarrow{I} \bar{x}$. By

$$\lim_{k \in I} \left\| \nabla f(x^k) \right\| = \lim_{k \in J} \left\| \nabla f(x^k) \right\| = 0$$

and the continuity of $\nabla f$, we get that $\nabla f(\bar{x}) = 0$ ensuring that $\bar{x}$ is a global minimizer of $f$ with the optimal value $f^* := f(\bar{x})$. Since the sequence $\left\{ f(x^k) \right\}$ is convergent and since $\bar{x}$ is an accumulation point of $\left\{ x^k \right\}$, we conclude that $f^* = f(\bar{x})$ is the limit of $\left\{ f(x^k) \right\}$. Now take any accumulation point $\widetilde{x}$ of $\left\{ x^k \right\}$ and find an infinite set $J' \subset \mathbb{N}$ with $x^k \xrightarrow{J'} \widetilde{x}$. As $\left\{ f(x^k) \right\}$ converges to $f^*$, we deduce that

$$f(\widetilde{x}) = \lim_{k \in J'} f(x^k) = \lim_{k \in \mathbb{N}} f(x^k) = f^*,$$

which implies that $\widetilde{x}$ is also a global minimizer of $f$. Assuming in addition that $f$ has a unique minimizer $\bar{x}$ and taking any accumulation point $\widetilde{x}$ of $\left\{ x^k \right\}$, we get that $\widetilde{x}$ is a minimizer of $f$, i.e., $\widetilde{x} = \bar{x}$. This means that $\bar{x}$ is the unique accumulation point of $\left\{ x^k \right\}$, and therefore $x^k \to \bar{x}$ as $k \to \infty$. $\qquad \square$

## C.2 Proof of Theorem 3.3

*Proof.* By (8), we find some $c_1 > 0, c_2 \in (0,1)$, and $K \in \mathbb{N}$ such that

$$\frac{1}{2}(2 - Lt_k - \varepsilon_k + Lt_k\varepsilon_k) \geq c_1, \quad \frac{1}{2}(1 - Lt_k) + \frac{Lt_k\varepsilon_k}{2} \leq c_2, \quad \text{and} \quad Lt_k < 1 \text{ for all } k \geq K.$$
(27)

Let us first verify the estimate

$$f(x^{k+1}) \leq f(x^k) - c_1 t_k \left\|\nabla f(x^k)\right\|^2 + c_2 t_k \varepsilon_k \text{ whenever } k \geq K.$$
(28)

To proceed, fix $k \in \mathbb{N}$ and deduce from the Cauchy-Schwarz inequality that

$$\begin{aligned}
\langle g^k, \nabla f(x^k)\rangle &= \langle g^k - \nabla f(x^k), \nabla f(x^k)\rangle + \left\|\nabla f(x^k)\right\|^2 \\
&\geq -\left\|g^k - \nabla f(x^k)\right\| \cdot \left\|\nabla f(x^k)\right\| + \left\|\nabla f(x^k)\right\|^2 \\
&\geq -\varepsilon_k \left\|\nabla f(x^k)\right\| + \left\|\nabla f(x^k)\right\|^2.
\end{aligned}$$
(29)

Since $\nabla f$ is Lipschitz continuous with constant $L$, it follows from the descent condition in (4) and the estimate (29) that

$$\begin{aligned}
f(x^{k+1}) &\leq f(x^k) + \left\langle \nabla f(x^k), x^{k+1} - x^k\right\rangle + \frac{L}{2}\left\|x^{k+1} - x^k\right\|^2 \\
&= f(x^k) - t_k\left\langle \nabla f(x^k), g^k\right\rangle + \frac{Lt_k^2}{2}\left\|g^k\right\|^2 \\
&= f(x^k) - t_k(1 - Lt_k)\left\langle \nabla f(x^k), g^k\right\rangle + \frac{Lt_k^2}{2}(\left\|g^k - \nabla f(x^k)\right\|^2 - \left\|\nabla f(x^k)\right\|^2) \\
&\leq f(x^k) - t_k(1 - Lt_k)\left(-\varepsilon_k\left\|\nabla f(x^k)\right\| + \left\|\nabla f(x^k)\right\|^2\right) + \frac{Lt_k^2\varepsilon_k^2}{2} - \frac{Lt_k^2}{2}\left\|\nabla f(x^k)\right\|^2 \\
&= f(x^k) - \frac{t_k}{2}(2 - Lt_k)\left\|\nabla f(x^k)\right\|^2 + t_k(1 - Lt_k)\varepsilon_k\left\|\nabla f(x^k)\right\| + \frac{Lt_k^2\varepsilon_k^2}{2} \\
&\leq f(x^k) - \frac{t_k}{2}(2 - Lt_k)\left\|\nabla f(x^k)\right\|^2 + \frac{1}{2}t_k(1 - Lt_k)\varepsilon_k\left(1 + \left\|\nabla f(x^k)\right\|^2\right) + \frac{Lt_k^2\varepsilon_k^2}{2} \\
&= f(x^k) - \frac{t_k}{2}(2 - Lt_k - \varepsilon_k + Lt_k\varepsilon_k)\left\|\nabla f(x^k)\right\|^2 + \frac{1}{2}t_k\varepsilon_k(1 - Lt_k) + \frac{Lt_k^2\varepsilon_k^2}{2} \\
&= f(x^k) - \frac{t_k}{2}(2 - Lt_k - \varepsilon_k + Lt_k\varepsilon_k)\left\|\nabla f(x^k)\right\|^2 + t_k\varepsilon_k\left(\frac{1}{2}(1 - Lt_k) + \frac{Lt_k\varepsilon_k}{2}.\right)
\end{aligned}$$

Combining this with (27) gives us (28). Defining $u_k := c_2 \sum_{i=k}^{\infty} t_i\varepsilon_i$ for $k \in \mathbb{N}$, we get that $u_k \to 0$ as $k \to \infty$ and $u_k - u_{k+1} = t_k\varepsilon_k$ for all $k \in \mathbb{N}$. Then (28) can be rewritten as

$$f(x^{k+1}) + u_{k+1} \leq f(x^k) + u_k - c_1 t_k\left\|\nabla f(x^k)\right\|^2, \quad k \geq K.$$
(30)

To proceed now with the proof of (i), we deduce from (30) combined with $\inf f(x^k) > -\infty$ and $u_k \to 0$ as $k \to \infty$ that

$$\begin{aligned}
c_1 \sum_{k=K}^{\infty} t_k\left\|\nabla f(x^k)\right\|^2 &\leq \sum_{k=K}^{\infty}\left(f(x^k) - f(x^{k+1}) + u_k - u_{k+1}\right) \\
&\leq f(x^K) - \inf_{k\in\mathbb{N}} f(x^k) + u_K < \infty.
\end{aligned}$$

Next we employ Lemma B.1 with $\alpha_k := \left\|\nabla f(x^k)\right\|$, $\beta_k := Lt_k$, and $\gamma_k := Lt_k\varepsilon_k$ for all $k \in \mathbb{N}$ to derive $\nabla f(x^k) \to 0$. Observe first that condition (a) is satisfied due to the the estimates

$$\begin{aligned}
\alpha_{k+1} - \alpha_k = \left\|\nabla f(x^{k+1})\right\| - \left\|\nabla f(x^k)\right\| &\leq \left\|\nabla f(x^{k+1}) - \nabla f(x^k)\right\| \\
&\leq L\left\|x^{k+1} - x^k\right\| = Lt_k\left\|g^k\right\| \\
&\leq Lt_k(\left\|\nabla f(x^k)\right\| + \left\|g^k - \nabla f(x^k)\right\|) \\
&\leq Lt_k(\left\|\nabla f(x^k)\right\| + \varepsilon_k) \\
&= \beta_k\alpha_k + \gamma_k \text{ for all } k \in \mathbb{N}.
\end{aligned}$$

Further, the conditions in (b) hold by (8) and $\sum_{k=1}^{\infty} t_k \left\| \nabla f(x^k) \right\|^2 < \infty$. As all the assumptions (a), (b) are satisfied, Lemma B.1 tells us that $\left\| \nabla f(x^k) \right\| = \alpha_k \to 0$ as $k \to \infty$.

To verify (ii), deduce from (30) that $\left\{ f(x^k) + u_k \right\}$ is nonincreasing. As $\inf_{k \in \mathbb{N}} f(x^k) > -\infty$ and $u_k \to 0$, we get that $\left\{ f(x^k) + u_k \right\}$ is bounded from below, and thus is convergent. Taking into account that $u_k \to 0$, it follows that $f(x^k)$ is convergent as well. Since $\bar{x}$ is an accumulation point of $\{x^k\}$, the continuity of $f$ tells us that $f(\bar{x})$ is also an accumulation point of $\{f(x^k)\}$, which immediately yields $f(x^k) \to f(\bar{x})$ due to the convergence of $\{f(x^k)\}$.

It remains to verify (iii). By the KL property of $f$ at $\bar{x}$, we find some $\eta > 0$, a neighborhood $U$ of $\bar{x}$, and a desingularizing concave continuous function $\varphi : [0, \eta) \to [0, \infty)$ such that $\varphi(0) = 0$, $\varphi$ is $C^1$-smooth on $(0, \eta)$, $\varphi' > 0$ on $(0, \eta)$, and we have for all $x \in U$ with $0 < f(x) - f(\bar{x}) < \eta$ that

$$\varphi'(f(x) - f(\bar{x})) \left\| \nabla f(x) \right\| \geq 1. \tag{31}$$

Let $\bar{K} > K$ be natural number such that $f(x^k) > f(\bar{x})$ for all $k \geq \bar{K}$. Define $\Delta_k := \varphi(f(x^k) - f(\bar{x}) + u_k)$ for all $k \geq \bar{K}$, and let $R > 0$ be such that $\mathbb{B}(\bar{x}, R) \subset U$. Taking the number $C$ from Assumption 2.3, remembering that $\bar{x}$ is an accumulation point of $\{x^k\}$, and using $f(x^k) + u_k \downarrow f(\bar{x})$, $\Delta_k \downarrow 0$ as $k \to \infty$ together with condition (9), we get by choosing a larger $\bar{K}$ that $f(x^{\bar{K}}) + u_{\bar{K}} < f(\bar{x}) + \eta$ and

$$\left\| x^{\bar{K}} - \bar{x} \right\| + \frac{1}{Cc_1} \Delta_{\bar{K}} + \sum_{k=\bar{K}}^{\infty} t_k \varphi' \left( \sum_{i=k}^{\infty} t_i \varepsilon_i \right)^{-1} + \sum_{k=\bar{K}}^{\infty} t_k \varepsilon_k < R. \tag{32}$$

Let us now show by induction that $x^k \in \mathbb{B}(\bar{x}, R)$ for all $k \geq \bar{K}$. The assertion obviously holds for $k = \bar{K}$ due to (32). Take some $\widehat{K} \geq \bar{K}$ and suppose that $x^k \in U$ for all $k = \bar{K}, \ldots, \widehat{K}$. We intend to show that $x^{\widehat{K}+1} \in \mathbb{B}(\bar{x}, R)$ as well. To proceed, fix some $k \in \left\{ \bar{K}, \ldots, \widehat{K} \right\}$ and get by $f(\bar{x}) < f(x^k) < f(x^k) + u_k < f(\bar{x}) + \eta$ that

$$\varphi'(f(x^k) - f(\bar{x})) \left\| \nabla f(x^k) \right\| \geq 1. \tag{33}$$

Combining this with $u_k > 0$ and $f(x^k) - f(\bar{x}) > 0$ gives us

$$\Delta_k - \Delta_{k+1} \geq \varphi'(f(x^k) - f(\bar{x}) + u_k)(f(x^k) + u_k - f(x^{k+1}) - u_{k+1}) \tag{34a}$$

$$\geq \varphi'(f(x^k) - f(\bar{x}) + u_k) c_1 t_k \left\| \nabla f(x^k) \right\|^2 \tag{34b}$$

$$\geq \frac{C}{(\varphi'(f(x^k) - f(\bar{x})))^{-1} + (\varphi'(u_k))^{-1}} c_1 t_k \left\| \nabla f(x^k) \right\|^2 \tag{34c}$$

$$\geq \frac{C}{\left\| \nabla f(x^k) \right\| + (\varphi'(u_k))^{-1}} c_1 t_k \left\| \nabla f(x^k) \right\|^2, \tag{34d}$$

where (34a) follows from the concavity of $\varphi$, (34b) follows from (30), (34c) follows from Assumption 2.3, and (34d) follows from (33). Taking the square root of both sides in (34d) and employing the AM-GM inequality yield

$$t_k \left\| \nabla f(x^k) \right\| = \sqrt{t_k} \cdot \sqrt{t_k \left\| \nabla f(x^k) \right\|^2} \leq \sqrt{\frac{1}{Cc_1} (\Delta_k - \Delta_{k+1}) t_k (\left\| \nabla f(x^k) \right\| + (\varphi'(u_k))^{-1})}$$

$$\leq \frac{1}{2Cc_1} (\Delta_k - \Delta_{k+1}) + \frac{1}{2} t_k \left( (\varphi'(u_k))^{-1} + \left\| \nabla f(x^k) \right\| \right). \tag{35}$$

Using the nonincreasing property of $\varphi'$ due to the concavity of $\varphi$ and the choice of $c_2 \in (0, 1)$ ensures that

$$(\varphi'(u_k))^{-1} = \left( \varphi'(c_2 \sum_{i=k}^{\infty} t_i \varepsilon_i) \right)^{-1} \leq \left( \varphi'(\sum_{i=k}^{\infty} t_i \varepsilon_i) \right)^{-1}.$$

Rearranging terms and taking the sum over $k = \bar{K}, \ldots, \widehat{K}$ of inequality (35) gives us

$$
\begin{aligned}
\sum_{k=\bar{K}}^{\widehat{K}} t_k \left\| \nabla f(x^k) \right\| &\leq \frac{1}{Cc_1} \sum_{k=\bar{K}}^{\widehat{K}} (\Delta_k - \Delta_{k+1}) + \sum_{k=\bar{K}}^{\widehat{K}} t_k \varphi'(u_k)^{-1} \\
&= \frac{1}{Cc_1} (\Delta_{\bar{K}} - \Delta_{\widehat{K}}) + \sum_{k=\bar{K}}^{\widehat{K}} t_k \varphi' \left( c_2 \sum_{i=k}^{\infty} t_i \varepsilon_i \right)^{-1} \\
&\leq \frac{1}{Cc_1} \Delta_{\bar{K}} + \sum_{k=\bar{K}}^{\widehat{K}} t_k \varphi' \left( \sum_{i=k}^{\infty} t_i \varepsilon_i \right)^{-1}.
\end{aligned}
$$

The latter estimate together with the triangle inequality and (32) tells us that

$$
\begin{aligned}
\left\| x^{\widehat{K}+1} - \bar{x} \right\| &= \left\| x^{\bar{K}} - \bar{x} \right\| + \sum_{k=\bar{K}}^{\widehat{K}} \left\| x^{k+1} - x^k \right\| \\
&= \left\| x^{\bar{K}} - \bar{x} \right\| + \sum_{k=\bar{K}}^{\widehat{K}} t_k \left\| g^k \right\| \\
&\leq \left\| x^{\bar{K}} - \bar{x} \right\| + \sum_{k=\bar{K}}^{\widehat{K}} t_k \left\| \nabla f(x^k) \right\| + \sum_{k=\bar{K}}^{\widehat{K}} t_k \left\| g^k - \nabla f(x^k) \right\| \\
&\leq \left\| x^{\bar{K}} - \bar{x} \right\| + \sum_{k=\bar{K}}^{\widehat{K}} t_k \left\| \nabla f(x^k) \right\| + \sum_{k=\bar{K}}^{\widehat{K}} t_k \varepsilon_k \\
&\leq \left\| x^{\bar{K}} - \bar{x} \right\| + \frac{1}{Cc_1} \Delta_{\bar{K}} + \sum_{k=\bar{K}}^{\infty} t_k \varphi' \left( \sum_{i=k}^{\infty} t_i \varepsilon_i \right)^{-1} + \sum_{k=\bar{K}}^{\infty} t_k \varepsilon_k < R.
\end{aligned}
$$

By induction, this means that $x^k \in \mathbb{B}(\bar{x}, R)$ for all $k \geq \bar{K}$. Then a similar device brings us to

$$
\sum_{k=\bar{K}}^{\widehat{K}} t_k \left\| \nabla f(x^k) \right\| \leq \frac{1}{Cc_1} \Delta_{\bar{K}} + \sum_{k=\bar{K}}^{\infty} t_k \varphi' \left( \sum_{i=k}^{\infty} t_i \varepsilon_i \right)^{-1} \quad \text{for all } \widehat{K} \geq \bar{K},
$$

which yields $\sum_{k=1}^{\infty} t_k \left\| \nabla f(x^k) \right\| < \infty$. Therefore,

$$
\begin{aligned}
\sum_{k=1}^{\infty} \left\| x^{k+1} - x^k \right\| &= \sum_{k=1}^{\infty} t_k \left\| g^k \right\| \leq \sum_{k=1}^{\infty} t_k \left\| \nabla f(x^k) \right\| + \sum_{k=1}^{\infty} t_k \left\| g^k - \nabla f(x^k) \right\| \\
&\leq \sum_{k=1}^{\infty} t_k \left\| \nabla f(x^k) \right\| + \sum_{k=1}^{\infty} t_k \varepsilon_k < \infty
\end{aligned}
$$

which justifies the convergence of $\left\{ x^k \right\}$ and thus completes the proof of the theorem. $\qquad \square$

## C.3 Proof of Theorem 4.1

*Proof.* Using $\left\| \nabla f(x^k) - g^k \right\| \leq \nu \left\| \nabla f(x^k) \right\|$ gives us the estimates

$$
\begin{aligned}
\left\| g^k \right\|^2 &= \left\| \nabla f(x^k) - g^k \right\|^2 - \left\| \nabla f(x^k) \right\|^2 + 2 \left\langle \nabla f(x^k), g^k \right\rangle \\
&\leq \nu^2 \left\| \nabla f(x^k) \right\|^2 - \left\| \nabla f(x^k) \right\|^2 + 2 \left\langle \nabla f(x^k), g^k \right\rangle \\
&= -(1 - \nu^2) \left\| \nabla f(x^k) \right\|^2 + 2 \left\langle \nabla f(x^k), g^k \right\rangle, \quad (36)
\end{aligned}
$$

$$
\begin{aligned}
\left\langle \nabla f(x^k), g^k \right\rangle &= \left\langle \nabla f(x^k), g^k - \nabla f(x^k) \right\rangle + \left\| \nabla f(x^k) \right\|^2 \\
&\leq \left\| \nabla f(x^k) \right\| \cdot \left\| g^k - \nabla f(x^k) \right\| + \left\| \nabla f(x^k) \right\|^2 \\
&\leq (1 + \nu) \left\| \nabla f(x^k) \right\|^2, \quad (37)
\end{aligned}
$$

$$
\begin{aligned}
- \left\langle \nabla f(x^k), g^k \right\rangle &= - \left\langle \nabla f(x^k), g^k - \nabla f(x^k) \right\rangle - \left\| \nabla f(x^k) \right\|^2 \\
&\leq \left\| \nabla f(x^k) \right\| \cdot \left\| g^k - \nabla f(x^k) \right\| - \left\| \nabla f(x^k) \right\|^2 \\
&\leq -(1 - \nu) \left\| \nabla f(x^k) \right\|^2, \quad (38)
\end{aligned}
$$

$$
\left\| \nabla f(x^k) \right\| - \left\| g^k - \nabla f(x^k) \right\| \leq \left\| g^k \right\| \leq \left\| \nabla f(x^k) \right\| + \left\| g^k - \nabla f(x^k) \right\|,
$$

which in turn imply that

$$
(1 - \nu) \left\| \nabla f(x^k) \right\| \leq \left\| g^k \right\| \leq (1 + \nu) \left\| \nabla f(x^k) \right\| \quad \text{for all } k \in \mathbb{N}. \quad (39)
$$

Using condition (12), we find $N \in \mathbb{N}$ so that $2 - 2\nu - Lt_k(1 + \nu)^2 \geq \delta$ for all $k \geq N$. Select such a natural number $k$ and use the Lipschitz continuity of $\nabla f$ with constant $L$ to deduce from the descent condition (4), the relationship $x^{k+1} = x^k - t_k g^k$, and the estimates (36)–(38) that

$$
\begin{aligned}
f(x^{k+1}) &\leq f(x^k) + \left\langle \nabla f(x^k), x^{k+1} - x^k \right\rangle + \frac{L}{2} \left\| x^{k+1} - x^k \right\|^2 \\
&= f(x^k) - t_k \left\langle \nabla f(x^k), g^k \right\rangle + \frac{Lt_k^2}{2} \left\| g^k \right\|^2 \\
&\leq f(x^k) - t_k \left\langle \nabla f(x^k), g^k \right\rangle + Lt_k^2 \left\langle \nabla f(x^k), g^k \right\rangle - \frac{Lt_k^2(1 - \nu^2)}{2} \left\| \nabla f(x^k) \right\|^2 \\
&\leq f(x^k) - t_k(1 - \nu) \left\| \nabla f(x^k) \right\|^2 + Lt_k^2(1 + \nu) \left\| \nabla f(x^k) \right\|^2 - \frac{Lt_k^2(1 - \nu^2)}{2} \left\| \nabla f(x^k) \right\|^2 \\
&= f(x^k) - \frac{t_k}{2} \left( 2 - 2\nu - Lt_k(1 + \nu)^2 \right) \left\| \nabla f(x^k) \right\|^2 \\
&\leq f(x^k) - \frac{\delta t_k}{2} \left\| \nabla f(x^k) \right\|^2 \quad \text{for all } k \geq N. \quad (40)
\end{aligned}
$$

It follows from the above that the sequence $\left\{ f(x^k) \right\}_{k \geq N}$ is nonincreasing, and hence the condition $\inf_{k \in \mathbb{N}} f(x^k) > -\infty$ ensures the convergence of $\left\{ f(x^k) \right\}$. This allows us to deduce from (40) that

$$
\frac{\delta}{2} \sum_{k=N}^{\infty} t_k \left\| \nabla f(x^k) \right\|^2 \leq \sum_{K=N}^{\infty} \left( f(x^k) - f(x^{k+1}) \right) \leq f(x^K) - \inf_{k \in \mathbb{N}} f(x^k) < \infty. \quad (41)
$$

Combining the latter with (39) and $x^{k+1} = x^k - t_k g^k$ gives us

$$
\sum_{k=1}^{\infty} \left\| x^{k+1} - x^k \right\| \cdot \left\| g^k \right\| = \sum_{k=1}^{\infty} t_k \left\| g^k \right\|^2 \leq (1 + \nu)^2 \sum_{k=1}^{\infty} t_k \left\| \nabla f(x^k) \right\|^2 < \infty. \quad (42)
$$

Now we are ready to verify all the assertions of the theorem. Let us start with (i) and show that $0$ in an accumulation point of $\left\{ g^k \right\}$. Indeed, supposing the contrary gives us $\varepsilon > 0$ and $K \in \mathbb{N}$ such that $\left\| g^k \right\| \geq \varepsilon$ for all $k \geq K$, and therefore

$$
\infty > \sum_{k=K}^{\infty} t_k \left\| g^k \right\|^2 \geq \sum_{k=K}^{\infty} t_k = \infty,
$$

which is a contradiction justifying that $0$ is an accumulation point of $\{g^k\}$. If $\bar{x}$ is an accumulation point of $\{x^k\}$, then by Lemma B.2 and (42), we find an infinite set $J \subset N$ such that $x^k \xrightarrow{J} \bar{x}$ and $g^k \xrightarrow{J} 0$. The latter being combined with (39) gives us $\nabla f(x^k) \xrightarrow{J} 0$, which yields the stationary condition $\nabla f(\bar{x}) = 0$.

To verity (ii), let $\bar{x}$ be an accumulation point of $\{x^k\}$ and find an infinite set $J \subset \mathbb{N}$ such that $x^k \xrightarrow{J} \bar{x}$. Combining this with the continuity of $f$ and the fact that $\{f(x^k)\}$ is convergent, we arrive at the equalities

$$f(\bar{x}) = \lim_{k \in J} f(x^k) = \lim_{k \in \mathbb{N}} f(x^k),$$

which therefore justify assertion (ii).

To proceed with the proof of the next assertion (iii), assume that $\nabla f$ is Lipschitz continuous with constant $L > 0$ and employ Lemma B.1 with $\alpha_k := \|\nabla f(x^k)\|$, $\beta_k := Lt_k(1 + \nu)$, and $\gamma_k := 0$ for all $k \in \mathbb{N}$ to derive that $\nabla f(x^k) \to 0$. Observe first that condition (a) of this lemma is satisfied due to the the estimates

$$\begin{aligned} \alpha_{k+1} - \alpha_k &= \|\nabla f(x^{k+1})\| - \|\nabla f(x^k)\| \\ &\leq \|\nabla f(x^{k+1}) - \nabla f(x^k)\| \leq L \|x^{k+1} - x^k\| \\ &= Lt_k \|g^k\| \leq Lt_k(1 + \nu) \|\nabla f(x^k)\| = \beta_k \alpha_k. \end{aligned}$$

The conditions in (b) of the lemma are satisfied since $\{t_k\}$ is bounded, $\sum_{k=1}^{\infty} t_k = \infty$ by (12), $\gamma_k = 0$, and

$$\sum_{k=1}^{\infty} \beta_k \alpha_k^2 = L(1 + \nu) \sum_{k=1}^{\infty} t_k \|\nabla f(x^k)\|^2 < \infty,$$

where the inequality follows from (41). Thus applying Lemma B.1 gives us $\nabla f(x^k) \to 0$ as $k \to \infty$.

To prove (iv), we verify the assumptions of Proposition B.3 for the sequences generated by Algorithm 2. It follows from (40) and $x^{k+1} = x^k - t_k g^k$ that

$$\begin{aligned} f(x^{k+1}) &\leq f(x^k) - \frac{\delta t_k}{2(1 + \nu)} \|\nabla f(x^k)\| \cdot \|g^k\| \\ &= f(x^k) - \frac{\delta}{2(1 + \nu)} \|\nabla f(x^k)\| \cdot \|x^{k+1} - x^k\|, \end{aligned} \tag{43}$$

which justify (H1) with $\sigma = \frac{\delta}{2(1+\nu)}$. Regarding condition (H2), assume that $f(x^{k+1}) = f(x^k)$ and get by (40) that $\nabla f(x^k) = 0$, which implies by $\|g^k - \nabla f(x^k)\| \leq \nu \|\nabla f(x^k)\|$ that $g^k = 0$. Combining this with $x^{k+1} = x^k - t_k g^k$ gives us $x^{k+1} = x^k$, which verifies (H2). Therefore, Proposition B.3 tells us that $\{x^k\}$ is convergent.

Let us now verify the final assertion (v) of the theorem. It is nothing to prove if $\{x^k\}$ stops at a stationary point after a finite number of iterations. Thus we assume that $\nabla f(x^k) \neq 0$ for all $k \in \mathbb{N}$. The assumptions in (v) give us $\bar{t} > 0$ and $N \in \mathbb{N}$ such that $t_k \geq \bar{t}$ for all $k \geq N$. Let us check that the assumptions of Proposition B.4 hold for the sequences generated by Algorithm 2 with $\tau_k := t_k$ and $d^k := -g^k$ for all $k \in \mathbb{N}$. The iterative procedure $x^{k+1} = x^k - t_k g^k$ can be rewritten as $x^{k+1} = x^k + t_k d^k$. Using the first condition in (39) and taking into account that $\nabla f(x^k) \neq 0$ for all $k \in \mathbb{N}$, we get that $g^k \neq 0$ for all $k \in \mathbb{N}$. Combining this with $x^{k+1} = x^k - t_k g^k$ and $t_k \geq \bar{t}$ for all $k \geq N$, tells us that $x^{k+1} \neq x^k$ for all $k \geq N$. It follows from (40) and (39) that

$$f(x^{k+1}) \leq f(x^k) - \frac{\delta t_k}{2(1 + \nu)^2} \|g^k\|^2. \tag{44}$$

This estimate together with the second inequality in (39) verifies (21) with $\beta = \frac{\delta}{2(1+\nu)^2}, \alpha = \frac{1}{1-\nu}$. As all the assumptions are verified, Proposition B.4 gives us the assertions:

- If $q \in (0, 1/2]$, then the sequence $\{x^k\}$ converges linearly to $\bar{x}$.

- If $q \in (1/2, 1)$, then we have the estimate

$$\left\| x^k - \bar{x} \right\| = \mathcal{O}\left( k^{-\frac{1-q}{2q-1}} \right).$$

The convergence rates of $\{f(x^k)\}$ and $\{\|\nabla f(x^k)\|\}$ follow now from Proposition B.5, and thus we are done. $\qquad\square$

### C.4 Proof of Theorem 4.2

*Proof.* Let $\{x^k\}$ be the sequence generated by Algorithm 2a. Defining $g^k := \nabla f(x^k + \rho_k \nabla f(x^k))$ and utilizing $\rho_k \leq \frac{\nu}{L}$, we obtain

$$\left\| g^k - \nabla f(x^k) \right\| = \left\| \nabla f(x^k + \rho_k \nabla f(x^k)) - \nabla f(x^k) \right\|$$
$$\leq L \left\| \rho_k \nabla f(x^k) \right\| \leq \nu \left\| \nabla f(x^k) \right\|,$$

which verifies the inexact condition in Step 2 of Algorithm 2. Therefore, all the convergence properties in Theorem 4.1 hold for Algorithm 2a. The proof for the convergence properties of Algorithm 2b can be conducted similarly. $\qquad\square$

### C.5 Proof of Corollary 3.5

*Proof.* Considering Algorithm 1a and defining $g^k = \nabla f\left( x^k + \rho_k \frac{d^k}{\|d^k\|} \right)$, we deduce that

$$\left\| g^k - \nabla f(x^k) \right\| \leq L \left\| x^k + \rho_k \frac{d^k}{\|d^k\|} - x^k \right\| = L \rho_k.$$

Therefore, Algorithm 1a is a specialization of Algorithm 1 with $\varepsilon_k = L\rho_k$. Combining this with (11) also gives us (8), thereby verifying all the assumptions in Theorem 3.3. Consequently, all the convergence properties outlined in Theorem 3.3 hold for Algorithm 1a. $\qquad\square$

## D Efficient normalized variants of SAM

In this section, we list several efficient normalized variants of SAM from [Foret et al., 2021, Liu et al., 2022, Li and Giannakis, 2023, Li et al., 2024] that are special cases of Algorithm 1a. As a consequence, all the convergence properties in Theorem 3.3 are satisfied for these methods.

---

**Algorithm 2c** [Foret et al., 2021] Sharpness-Aware Minimization (SAM)

---

*Step* 0. Choose $x^1 \in \mathbb{R}^n$, $\{\rho_k\} \subset [0, \infty)$, and $\{t_k\} \subset [0, \infty)$. For $k = 1, 2, \ldots$, do the following:

*Step* 1. Set $x^{k+1} = x^k - t_k \nabla f\left( x^k + \rho_k \frac{\nabla f(x^k)}{\|\nabla f(x^k)\|} \right)$.

---

**Algorithm 2d** [Liu et al., 2022] Random Sharpness-Aware Minimization (RSAM)

---

*Step* 0. Choose $x^1 \in \mathbb{R}^n$, $\{\rho_k\} \subset [0, \infty)$, and $\{t_k\} \subset [0, \infty)$. For $k = 1, 2, \ldots$, do the following:

*Step* 1. Construct a random vector $\Delta^k \in \mathbb{R}^n$ and set $g^k = \nabla f(x^k + \Delta^k)$.

*Step* 2. Set $x^{k+1} = x^k - t_k \nabla f\left( x^k + \rho_k \frac{\Delta^k + \lambda g^k}{\|\Delta^k + \lambda g^k\|} \right)$.

---

**Algorithm 2e** [Li and Giannakis, 2023] Variance suppressed sharpness aware optimization (VaSSO)

---

*Step* 0. Choose $x^1 \in \mathbb{R}^n$, $d^1 \in \mathbb{R}^n$, $\{\rho_k\} \subset [0, \infty)$, $\{t_k\} \subset [0, \infty)$, $\theta \in (0, 1)$. For $k \geq 1$, do the following:

*Step* 1. Set $d^k = (1 - \theta)d^{k-1} + \theta \nabla f(x^k)$.

*Step* 2. Set $x^{k+1} = x^k - t_k \nabla f\left( x^k + \rho_k \frac{d^k}{\|d^k\|} \right)$.

---

**Algorithm 2f** [Li et al., 2024] Friendly Sharpness-Aware Minimization (F-SAM)

---

*Step* 0. Choose $x^1 \in \mathbb{R}^n$, $d^1 \in \mathbb{R}^n$, $m^1 \in \mathbb{R}^n$, $\sigma \in \mathbb{R}$, $\{\rho_k\} \subset [0, \infty)$, $\{t_k\} \subset [0, \infty)$, $\theta > 0$. For $k \geq 1$:

*Step* 1. Set $m^k = (1 - \theta)m^{k-1} + \theta \nabla f(x^k)$.

*Step* 2. Set $d^k = \nabla f(x^k) - \sigma m^k$.

*Step* 3. Set $x^{k+1} = x^k - t_k \nabla f\left(x^k + \rho_k \frac{d^k}{\|d^k\|}\right)$.

---

*Remark* D.1. It is clear that Algorithms 2c-2f are specializations of Algorithm 1a with $d^k = \nabla f(x^k)$ in Algorithm 2c, $d^k = \Delta^k + \lambda g^k$ in Algorithm 2d, and $d^k$ constructed inductively for Algorithm 2e and Algorithm 2f.

# E Numerical experiments on SAM constant and SAM almost constant

In this section, we present numerical results to support our claim in Remark 3.6 that SAM with an almost constant perturbation radius $\rho_k = \frac{C}{k^p}$ for $p$ close to 0, e.g., $p = 0.001$, generates similar results to SAM with a constant perturbation radius $\rho = C$. To do so, we consider the function $f(x) = \sum_{i=1}^n \log(1 + (Ax - b)_i^2)$, where $A$ is an $n \times n$ matrix, and $b$ is a vector in $\mathbb{R}^n$. In the experiment, we construct $A$ and $b$ randomly with $n \in 2, 20, 50, 100$. The methods considered in the experiment are GD with a diminishing step size, SAM with a diminishing step size and a constant perturbation radius of $0.1$, and lastly, SAM with a diminishing step size and a variable radius $\rho_k = \frac{C}{k^p}$, for $p \in 1, 0.1, 0.001$. We refer to the case $p = 0.001$ as the "almost constant" case, as $\rho_k = \frac{C}{k^p}$ is numerically similar to $C$ when we consider a small number of iterations. The diminishing step size is chosen as $t_k = (0.1/n)/k$ at the $k^{\text{th}}$ iteration, where $n$ is the dimension of the problem. To make the plots clearer, we choose the initial point $x^1$ near the solution, which is $x^1 = x^\infty + (0.1/n^2)\mathbf{1}_n$, where $x^\infty$ is a solution of $Ax = b$, and $\mathbf{1}_n$ is the all-ones vector in $\mathbb{R}^n$. All the algorithms are executed for $100n$ iterations. The results presented in Figure 3 show that SAM with a constant perturbation and SAM with an almost constant perturbation have the same behavior regardless of the dimension of the problem. This is simply because $\frac{C}{k^{0.001}}$ is almost the same as $C$. This also tells us that the convergence rate of these two versions of SAM is similar. Since SAM with a constant perturbation radius is always preferable in practice [Foret et al., 2021, Dai et al., 2023], this highlights the practicality of our development.

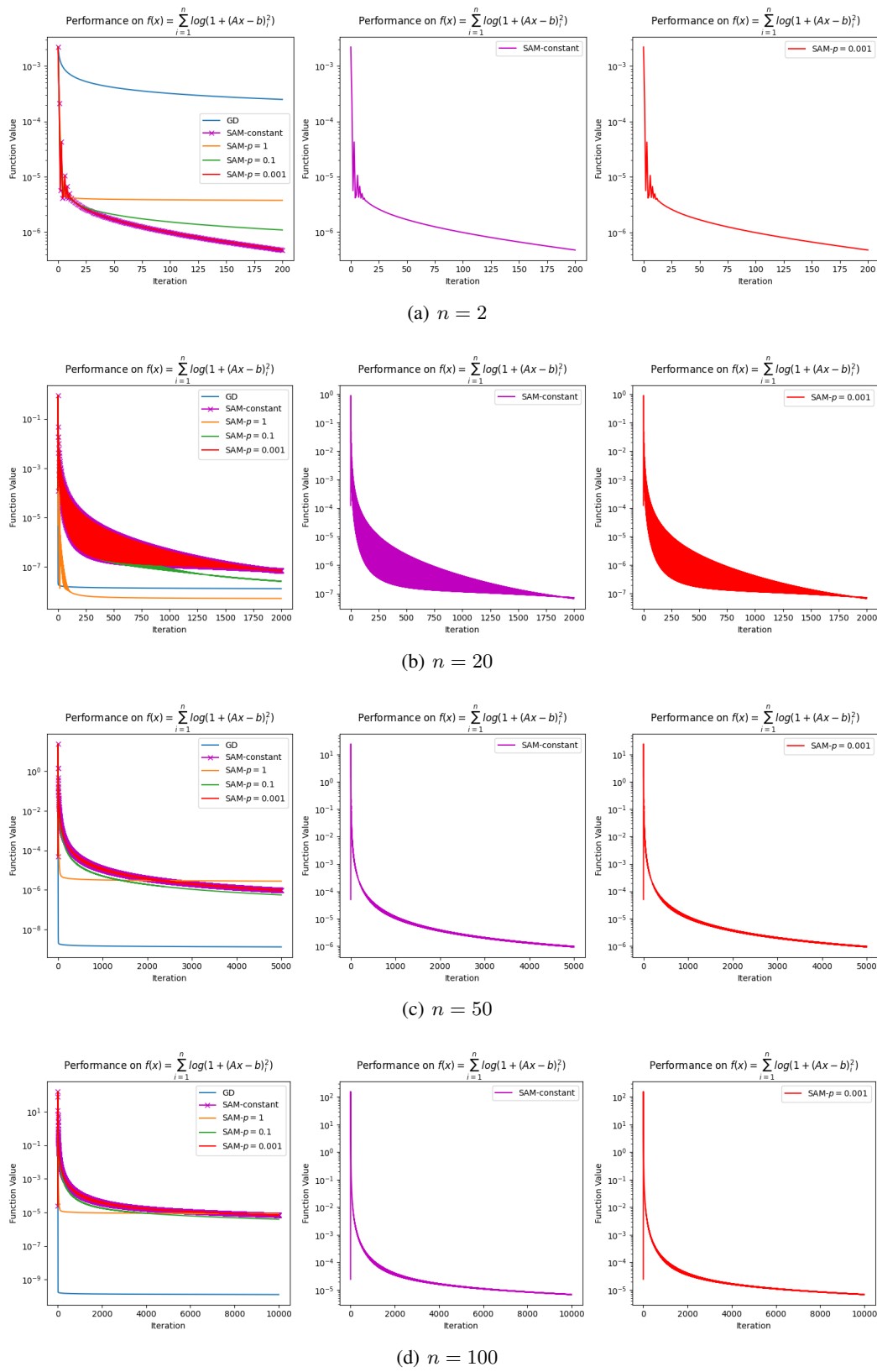

Figure 3: SAM with constant perturbation and SAM almost constant perturbation

# F Numerical experiments on SAM with SGD without momentum as base optimizer

**CIFAR-10, CIFAR-100, and Tiny ImageNet**. The training configurations for these datasets follow a similar structure to Section 5, excluding momentum, which we set to zero. The results in Table 5 report test accuracy on CIFAR-10 and CIFAR-100. Table 6 shows the performance of SAM on momentum and without momentum settings. Each experiment is run once, and the highest accuracy for each column is highlighted in bold.

Table 5: Additional Numerical Results on CIFAR-10, CIFAR-100 for SAM without momentum

| | CIFAR-10 | | | CIFAR-100 | | |
|---|---|---|---|---|---|---|
| Model | ResNet18 | ResNet34 | WRN28-10 | ResNet18 | ResNet34 | WRN28-10 |
| Constant | 93.64 | 94.26 | 93.04 | 72.07 | 72.57 | 71.11 |
| Diminish 1 | 88.87 | 89.79 | 86.81 | 65.99 | 67.04 | 51.31 |
| Diminish 2 | **94.56** | **94.44** | **93.66** | **74.24** | **74.95** | **74.23** |
| Diminish 3 | 90.84 | 91.23 | 88.70 | 69.69 | 70.54 | 60.64 |

Table 6: Additional Numerical Results on Tiny ImageNet [Le and Yang, 2015] for SAM with and without momentum

| Tiny ImageNet | Momentum | | | Without Momentum | |
|---|---|---|---|---|---|
| Model | ResNet18 | ResNet34 | WRN28-10 | ResNet18 | ResNet34 |
| Constant | 48.58 | 48.34 | 53.34 | 54.90 | 57.36 |
| Diminish 1 | 50.36 | 51.24 | 58.37 | 55.82 | 55.96 |
| Diminish 2 | 48.70 | 49.06 | 52.74 | 57.30 | **60.00** |
| Diminish 3 | **51.46** | **53.98** | **58.68** | **57.86** | 57.82 |

# G Additional Remarks

*Remark* G.1. Assumption 2.3 is satisfied with constant $C = 1$ for $\varphi(t) = Mt^{1-q}$ with $M > 0$ and $q \in [0, 1)$. Indeed, taking any $x, y > 0$ with $x + y < \eta$, we deduce that $(x + y)^q \leq x^q + y^q$, and hence

$$[\varphi'(x+y)]^{-1} = \frac{1}{M(1-q)}(x+y)^q \leq \frac{1}{M(1-q)}(x^q + y^q) = (\varphi'(x))^{-1} + (\varphi'(y))^{-1}.$$

*Remark* G.2. Construct an example to demonstrate that the conditions in (11) do not require that $\rho_k$ converges to 0. Let $L > 0$ be a Lipschitz constant of $\nabla f$, let $C$ be a positive constant such that $C < 2/L$, let $P \subset \mathbb{N}$ be the set of all perfect squares, let $t_k = \frac{1}{k}$ for all $k \in \mathbb{N}, p > 0$, and let $\{\rho_k\}$ be constructed as follows:

$$\rho_k = \begin{cases} C, & k \in P, \\ \frac{C}{k^p}, & k \notin P, \end{cases} \quad \text{which yields} \quad t_k\rho_k = \begin{cases} \frac{C}{k}, & k \in P, \\ \frac{C}{k^{p+1}}, & k \notin P. \end{cases}$$

It is clear from the construction of $\{\rho_k\}$ that $\limsup_{k\to\infty} \rho_k = C > 0$, which implies that $\{\rho_k\}$ does not convergence to 0. We also immediately deduce that $\sum_{k=1}^{\infty} t_k = \infty$, $t_k \downarrow 0$, and $\limsup \rho_k = C < \frac{2}{L}$, which verifies the first three conditions in (11). The last condition in (11) follows from the estimates

$$\sum_{k=1}^{\infty} t_k\rho_k = \sum_{k \in P} t_k\rho_k + \sum_{k \notin P} t_k\rho_k \leq \sum_{k \in P} \frac{C}{k} + \sum_{k \in \mathbb{N}} \frac{C}{k^{p+1}}$$

$$= \sum_{k \in \mathbb{N}} \frac{C}{k^2} + \sum_{k \in \mathbb{N}} \frac{C}{k^{p+1}} < \infty.$$

*Remark* G.3 (on Assumption (9)). Supposing that $\varphi(t) = Mt^{1-q}$ with $M > 0, q \in (0, 1)$ and letting $C = 1/(M(1-q))$, we get that $(\varphi'(t))^{-1} = Ct^q$ for $t > 0$ is an increasing function. If $t_k := \frac{1}{k}$ and $\varepsilon_k := \frac{1}{k^p}$ with $p > 0$, we have

$$\sum_{i=k}^{\infty} t_i \varepsilon_i = \sum_{i=k}^{\infty} \frac{1}{i^{1+p}} \leq \int_k^{\infty} \frac{1}{x^{1+p}} dx = -\frac{1}{px^p}\Big|_k^{\infty} = \frac{1}{pk^p},$$

which yields the relationships

$$\left(\varphi'\left(\sum_{i=k}^{\infty} t_i \varepsilon_i\right)\right)^{-1} \leq \left(\varphi'\left(\frac{1}{p(k+1)^p}\right)\right)^{-1} = \frac{C}{p^q k^{pq}}.$$

Therefore, we arrive at the claimed conditions

$$\sum_{k=1}^{\infty} t_k \left(\varphi'\left(\sum_{i=k}^{\infty} t_i \varepsilon_i\right)\right)^{-1} \leq \sum_{k=1}^{\infty} \frac{1}{k} \frac{C}{p^q k^{pq}} = \sum_{k=1}^{\infty} \frac{C}{p^q k^{1+pq}} < \infty.$$

*Remark* G.4. Let us finally compare the results presented in Theorem 4.1 with that in Andriushchenko and Flammarion [2022]. All the convergence properties in Andriushchenko and Flammarion [2022] are considered for the class of $\mathcal{C}^{1,L}$ functions, which is more narrow than the class of $L$-descent functions examined in Theorem 4.1(i). Under the convexity of the objective function, the convergence of the sequences of the function values at *averages of iteration* is established in [Andriushchenko and Flammarion, 2022, Theorem 11], which does not yield the convergence of either the function values, or the iterates, or the corresponding gradients. In the nonconvex case, we derive the stationarity of accumulation points, the convergence of the function value sequence, and the convergence of the gradient sequence in Theorem 4.1. Under the strong convexity of the objective function, the linear convergence of the sequence of iterate values is established Andriushchenko and Flammarion [2022, Theorem 11]. On the other hand, our Theorem 4.1 derives the convergence rates for the sequence of iterates, sequence of function values, and sequence of gradient under the KL property only, which covers many classes of nonconvex functions. Our convergence results address variable stepsizes and bounded radii, which also cover the case of constant stepsize and constant radii considered in Andriushchenko and Flammarion [2022].

