# OpenReview forum: "Fundamental Convergence Analysis of Sharpness-Aware Minimization"
_NeurIPS.cc/2024/Conference — NeurIPS 2024 poster_

### Official Review · Reviewer_cPcD · 2024-06-26

**Soundness:** 3
**Presentation:** 3
**Contribution:** 3
**Rating:** 6
**Confidence:** 3

**Summary:**

The authors provide a fundamental convergence analysis of sharpness-aware minimization algorithms,
including the normalized and unnormalized variants.
The analysis of normalized variants is based on the theoretical results of inexact gradient descent methods,
while the analysis of unnormalized variants is based on the inexact gradient descent method with relative errors.
Fundamental convergence properties have been established for SAM algorithms in both convex and non-convex cases.

**Strengths:**

- SAM is an optimizer widely used in searching for flat minima, thus, studying its convergence is an important problem.
- The paper establishes theoretical convergence for both normalized and unnormalized SAM algorithms, while most existing convergence results are only for unnormalized variants. As the normalized SAM is more standard than the unnormalized SAM, establishing convergence for normalized SAM is a key contribution.
- Compared with existing analysis on SAM, many new fundamental properties are established in Sec 3.2 and Sec.4
- the paper is well-written, easy to follow

**Weaknesses:**

- I think the analysis of the convex case (Sec 3.1) is meaningless. As for the convex function, usually, we can use some convex programming methods to obtain the global optimal solution.
- If Theorem 3.3 and Theorem 3.4 are from existing works, some citations for them are necessary (e.g., [Khanh et al., 2023a,b,c, 2024])
- In Corollary 3.5, the assumption on $\rho_t$ is not standard. In practice, $\rho_t$ is a constant, e.g., 0.05
- The convergence analysis of SAM in this paper cannot provide some theoretical insights on the advantage of SAM over ERM.

**Questions:**

- the results reported in Table 4 are much lower than other publications. e.g.,
  - resnet18, cifar10, 94.10 vs 96.52 (https://arxiv.org/pdf/2110.03141)
  - resnet18, cifar100, 71.77 vs 80.17 (https://arxiv.org/pdf/2110.03141)

---

> ### Author Rebuttal · Authors · 2024-08-05
>
> Dear Reviewer cPcD,
>
> Thank you for your precious time reviewing our paper. We answer your question as follows
>
> *1. I think the analysis of the convex case (Sec 3.1) is meaningless. As for the convex function, usually, we can use some convex programming methods to obtain the global optimal solution.*
>
> We agree that SAM is not designed for convex functions. However, for the completeness of the analysis, and motivated by the fact that convex functions (even quadratic functions) have been considered in recent studies (Bartlett et al., 2023; Ahn et al., 2023), we presented the analysis for the convex case.
>
> *2. If Theorem 3.3 and Theorem 3.4 are from existing works, some citations for them are necessary (e.g., [Khanh et al., 2023a,b,c, 2024])*
>
> The proof of Theorem 3.3 is completely new and not deduced from [Khanh et al., 2023a,b,c, 2024]. Could you please check the location of Theorem 3.4 again, as we only have Example 3.4?
>
> *3. In Corollary 3.5, the assumption on rho is not standard. In practice, rho is a constant, e.g., 0.05.*
>
> We agree that in practice rho is a constant as you mentioned. In our analysis, it is required that when rho is almost constant,  e.g., 0.05/(k^0.00001), with k being the number of iteration. This is, in fact, aligned with the practical use of rho, as  0.05~0.05/(k^0.00001). This phenomenon is also numerically tested in Appendix E, and thus consequently explains why SAM seems to converge to a stationary point in practice.
>
> *4. The results reported in Table 4 are much lower than other publications.*
>
>  This is because we use **diminishing stepsize and constant stepsize** to demonstrate our **theory**. In practice, **consine stepsize scheduler** is usually employed to achieve the best performance. We unfortunately could not process the analysis using this type of stepsize and will consider it in the future development.
>
> Best regards,
>
> Authors

---

> > ### Comment · Reviewer_cPcD · 2024-08-12
> > **reply to the rebuttal**
> >
> > I appreciate the authors' response.
> > The first three concerns are resolved.
> > As this is a theoretical paper, I think the 4th concern about experiments is not very crucial.
> > Hence, I will raise my score to 6 to reflect it.

---

### Official Review · Reviewer_ZtLt · 2024-07-06

**Soundness:** 3
**Presentation:** 3
**Contribution:** 2
**Rating:** 4
**Confidence:** 4

**Summary:**

This paper studies the convergence properties of Sharpness-Aware Minimization (SAM) and some of its variants. The authors use the inexact gradient descent to present convergence guarantees for convex and non-convex cases.

**Strengths:**

I think it is interesting to unify different variants of SAM under the inexact gradient framework.

**Weaknesses:**

- The results do not say much about the quality of the solution in the non-convex case. I think this makes the results less interesting in practice.

- The results seem rather a straightforward application of prior work. I think the contribution of the work is limited.

- The results do not discuss the stochastic setting.

I think based on these points, the results are rather limited and I don't think the paper comes up with a new proof technique.

**Questions:**

NA

---

> ### Author Rebuttal · Authors · 2024-08-05
>
> Dear Reviewer ZtLt,
>
> We really appreciate your precious time reviewing our paper. However, we hardly agree with most of your opinions on the weakness of the paper.
>
> *1. The results seem rather a straightforward application of prior work. I think the contribution of the work is limited.*
>
> In this paper, we develop a version of IGD with absolute error, which can cover variants of SAM. This version of IGD has **not been considered** in previous works, and the convergence analysis of this version is **significantly more complex**. This complexity is because the function value sequence generated by IGD with absolute error may not be decreasing, which makes all the convergence framework for monotone sequence before broken.This main contribution, in addition to minor ones for unnormalized version of SAM is not straightforward, and is significant.
>
> *2. The results do not say much about the quality of the solution in the non-convex case. I think this makes the results less interesting in practice. The results do not discuss the stochastic setting.*
>
> As introduced, we only focus on the fundamental convergence properties of SAM and its variants. Therefore, it is reasonable that we do not focus on the quality of the solution, given that the **ideas and techniques** in the convergence analysis are **significantly novel**.
>
> *3. I don't think the paper comes up with a new proof technique.*
>
> **The techniques we used in this paper are new**. This is discussed in the answer 1 above.
>
> We would appreciate it if you could reexamine the paper and consider raising your score based on the contributions of our work.
>
> Best regards,
>
> Authors

---

> > ### Comment · Reviewer_ZtLt · 2024-08-14
> >
> > I will increase my score to 4, but I don't think my concerns regarding the limitations of the paper are addressed.

---

> > > ### Author Response · Authors · 2024-08-14
> > >
> > > Dear Reviewer ZtLt,
> > >
> > > Thank you very much for reexamining our paper.
> > >
> > > Best regards,
> > >
> > > Authors

---

### Official Review · Reviewer_Yksm · 2024-07-09

**Soundness:** 3
**Presentation:** 3
**Contribution:** 3
**Rating:** 6
**Confidence:** 4

**Summary:**

In this paper, the authors provide a comprehensive understanding of the SAM and its variants. They establish the convergence for SAM under different settings. Unlike the convergence results in the literature, they establish the last-iterate convergence rather than the best-iterate convergence or function value convergence for SAM. By regarding SAM as an inexact gradient descent method (IGD), their analysis can also be applied to other variants.

**Strengths:**

* They establish the fundamental convergence properties for SAM and its variants.

* Their analysis is universal and can be applied to other versions of SAM.

* The paper is well-written and easy to follow.

**Weaknesses:**

* In line 480, you should refer to Proposition 3.9 in Khanh et al.

* Without the KL assumption, would it be possible to derive the convergence rates for SAM based on your analysis?

* The main idea is to regard the SAM as a special case of IGD. Would you please elaborate more on your theoretical contribution compared with IGD?

**Questions:**

See the Weakness part.

**Limitations:**

They have mentioned some in their paper.

---

> ### Author Rebuttal · Authors · 2024-08-05
>
> Dear Reviewer Yksm,
>
> Thank you for your precious time reviewing our paper. We answer your question as follows.
>
> *1. In line 480, you should refer to Proposition 3.9 in Khanh et al.*
>
>  It is actually Proposition 2.4 in the published version of the paper, as we cited.
>
> *2. Without the KL assumption, would it be possible to derive the convergence rates for SAM based on your analysis?*
>
> Without the KL assumption, we don't know how to derive the convergence rates for SAM or its variants. This is mainly because of the generality of the KL property, which is satisfied for almost all objective functions in practice. We may consider using other types of regularity properties, such as the metric subregularity of the gradient mapping. However, this property is again equivalent to the KL property in most circumstances.
>
> *3. The main idea is to regard the SAM as a special case of IGD. Would you please elaborate more on your theoretical contribution compared with IGD?*
>
> In this paper, we develop a version of IGD with absolute error, which can cover variants of SAM. This version of IGD **has not been considered in previous works**, and the convergence analysis of this version is **significantly more complex**. This complexity is because the function value sequence generated by IGD with absolute error may not be decreasing, which makes all the convergence framework for monotone sequence before broken. This is the main contribution of our paper, both in terms of the idea and technical aspects.
>
> Best regards,
>
> Authors

---

> > ### Comment · Reviewer_Yksm · 2024-08-14
> >
> > I thank the authors for addressing my questions. I will maintain my score. While I find the perspective of considering SAM as IGD interesting, I believe the paper should include more discussion on your technical contributions.

---

> > > ### Author Response · Authors · 2024-08-14
> > >
> > > Dear Reviewer Yksm,
> > >
> > > Thank you very much. We will include more discussion of the technical contributions in the introduction, following your suggestions.
> > >
> > > Best regards,
> > >
> > > Authors

---

### Official Review · Reviewer_wdGg · 2024-07-24

**Soundness:** 3
**Presentation:** 2
**Contribution:** 3
**Rating:** 6
**Confidence:** 5

**Summary:**

### Summary:

This paper studies various notions of convergence for SAM. The results are summarized in Table 3. Indeed, in Theorem 3.2, they prove the convergence of SAM for smooth convex function for several definitions under diminishing stepsize. Moreover, in Figure 1, they argue that SAM cannot converge with constant stepsize. In Theorem 3.3 they use Inexact Gradient Descent (IGD) in Algorithm 1 to prove convergence results in the non-convex setting, and they generalize it to unnormalized gradient methods like U-SAM in Theorem 4.1. The paper concludes with experiments.

**Strengths:**

### Pros:

- study of convergence of SAM is an important problem in optimization
- well citation to previous related works

**Weaknesses:**

### Cons:

- the paper is not well-written
- the method for non-convex convergence (IGD) is not well motivated and it seems too general to be used for the study of SAM

**Questions:**

### Questions/Comments:

This is an interesting paper about the theoretical understanding of SAM convergence. However, I think this paper should be majorly revised. The authors introduce a lot of notions and formulae in the first pages and later without accurately explaining them in the paper. I think because this paper could have general audience, it should be understandable to most part of the community. I recommend authors to move a lot of detailed formulae in the main body of the paper and explain necessary definitions/ideas in words through the paper. The writing can be improved for the next version of the paper.

Moreover, the use of IGD seems too general for proving SAM convergence. The authors fail to well motivate using IGDs, as well as how they are enough for the study of SAM. In other words, SAM is way more structured than IGDs and the considered framework seems too general. More explanations are expected for this.

- line 42 -- typo

---

> ### Author Rebuttal · Authors · 2024-08-05
>
> Dear Reviewer wdGg,
>
> We really appreciate your precious time reviewing our paper. However, we hardly agree with your opinions on the paper.
>
> *1. The paper is not well-written.*
>
> This comment is going against with opinions of other reviewers, as all of them they view the presentation of our paper in a good way (3/4 points).
>
> *2. IGD is not well motivated and it seems too general to be used for the study of SAM*
>
> Firstly, the motivation for using IGD stems from the construction of SAM, which is already demonstrated by a one-line proof within our results (Appendix C5). Although IGD is more general, to understand the convergence properties of SAM alone, it is sufficient. In addition, we successfully proved the convergence properties of SAM using IGD, which further emphasizes the validity of our approach.
>
> We would appreciate it if you could reexamine the paper and consider raising your score based on the contributions of our work.
>
> Best regards,
>
> Authors

---

> > ### Comment · Reviewer_wdGg · 2024-08-12
> >
> > Dear authors, thank you for your response. I believe your work is indeed interesting and influential, and my concerns are mainly about its presentation. The use of IGD is less motivated, and the paper is less accessible to a broad audience at this conference. However, to acknowledge the value of your work, I have decided to slightly increase my initial score. I strongly recommend making significant revisions so that the message of the paper is clearer to a broader audience.

---

> > > ### Author Response · Authors · 2024-08-13
> > >
> > > Dear Reviewer wdGg,
> > >
> > > Thank you very much for reexamining our paper. We will make revisions to the paper following your suggestions.
> > >
> > > Best regards,
> > >
> > > Authors

---

### Author Rebuttal · Authors · 2024-08-05

Dear Reviewers,

Thank you all for the valuable feedback. We discuss questions and concerns raised in the answers below. If you have any further questions, please do not hesitate to contact us. We are more than happy to answer them!

Best regards,

Authors

---

### Comment · Area_Chair_4KW5 · 2024-08-14
**Discussions**

Dear reviewers,

Thanks very much for your great efforts in the review process. Please read the authors' response and confirm it.  You are encouraged to discuss with all of us if you have any concerns.

Thanks,
AC

---

### Decision · Program_Chairs · 2024-09-25

**Decision:**

Accept (poster)

**Comment:**

Most of the reviewers think the paper is interesting.  There seems to be something new in the proof.  I recommend an acceptance.